# Seasonal variability of ocean heat transport and ice shelf basal melt around Antarctica

Fabio Boeira Dias<sup>1,2</sup>, Matthew H. England<sup>1,2,\*</sup>, Adele K. Morrison<sup>3,\*</sup>, and Benjamin K. Galton-Fenzi<sup>4,5,2,\*</sup>

**Correspondence:** Fabio Boeira Dias (f.boeira dias@unsw.edu.au)

Abstract. The delivery of ocean heat to Antarctic ice shelves is due to intrusions of waters warmer than the local freezing point temperature. Changes in the supply of ocean heat will determine how rapidly ice shelves melt at their base, which affects Antarctic Ice Sheet mass loss and future global mean sea level rise. However, processes driving ice shelf basal melting are still poorly understood. Here we investigate the drivers of heat convergence along the Antarctic margins by performing an ocean heat budget analysis using a high-fidelity 4 km circum-Antarctic ocean—ice-shelf model. The simulation produces high basal melting in West Antarctica associated with sustained ocean heat convergence driven by advection of relatively warm deep water intrusions, with minimal seasonality in both heat supply and basal melting. For East Antarctica, ice shelves have substantial basal melt seasonality, driven by strong air-sea winter cooling over the continental shelf depressing shallow melting, while in summer, increased heat inflow towards the ice shelves is driven by surface-warmed waters that subduct under shallow regions of ice, increasing melt. The high seasonality of basal melting in East Antarctic ice shelves is responsive to interactions between the atmospheric forcing, the local icescape, and the activity of coastal polynyas. Our results suggest that seasonal changes in future climate change scenarios are critical in determining the duration and intensity of air-sea fluxes with substantial impacts on ice shelf basal melting and ice sheet and sea level budgets.

# 1 Introduction

Ocean heat delivery towards Antarctica is a key driver of ice shelf basal melting, where changes in heat transport can drive ice shelf thinning and grounding line retreat, with implications for ice sheet stability and future global sea-level changes. The stability of the Antarctic Ice Sheet, which holds the equivalent of 58 m of global mean sea-level rise (Fretwell et al., 2013), depends on the buttressing effect from floating ice shelves that impedes ice flow towards the ocean (Joughin et al., 2010). Satellite observations show that the Antarctic Ice Sheet lost mass over the last few decades (Adusumilli et al., 2020), of which about half is due to ocean-driven basal melting of the ice shelves (Rignot et al., 2013; Pritchard et al., 2012), highlighting the

<sup>&</sup>lt;sup>1</sup>Centre of Marine Science and Innovation, University of New South Wales, Sydney NSW, Australia

<sup>&</sup>lt;sup>2</sup>Australian Centre for Excellence in Antarctic Science, Hobart/Nipaluna TAS, Australia

<sup>&</sup>lt;sup>3</sup>Research School of Earth Sciences, Australian National University, Canberra ACT, Australia

<sup>&</sup>lt;sup>4</sup>Australian Antarctic Division, Hobart/Nipaluna TAS, Australia

<sup>&</sup>lt;sup>5</sup>The Australian Antarctic Program Partnership, Institute for Marine and Antarctic Studies, Hobart/Nipaluna TAS, Australia

<sup>\*</sup>These authors contributed equally to this work.

importance of the Antarctic coastal ocean circulation and dynamics for sea-level projections.

The latest sea-level projections relied on climate model outputs to force ice sheet models under ISMIP6 (Edwards et al., 2021; Seroussi et al., 2024). However, the ocean component of CMIP5 and CMIP6 models have often coarse (1°) horizontal resolution (~55 km at 60°S), inherent biases in continental shelf temperatures (Purich and England, 2021), and lack active ice-shelf and ice-sheet coupling (Fox-Kemper et al., 2023). At 1° horizontal resolution, eddies, tides and other small-scale processes at the continental shelf (Stewart et al., 2018; Jourdain et al., 2019; Hausmann et al., 2020), such as Antarctic Bottom Water formation (Heuzé, 2021) and channelised flow associated with bathymetric troughs (Drews, 2015; Alley et al., 2016; Marsh et al., 2016; Milillo et al., 2019) are not resolved. The lack of ocean–ice-shelf–ice sheet coupling means that feedbacks between the ocean-atmosphere-cryosphere system are also not properly resolved; nor are some essential features of the icescape, such as landfast sea ice (Fraser et al., 2023), icebergs (Merino et al., 2016) and the ice shelves. The Antarctic Ice Sheet contribution to CMIP6 sea-level projection has the largest uncertainty among all contributors to global mean sea-level changes (Fox-Kemper et al., 2023). The leading sources of the uncertainty result from the choice of ice sheet models, followed by uncertainties in ocean-ice interactions and the choice of climate model (Seroussi et al., 2023), pointing to the critical role of limitations in the ocean forcing.

The ocean role in the basal melting of the Antarctic Ice Shelves is a important contributor to the uncertainties in the Antarctic Ice Sheet projections. The ocean heat delivered to the ocean—ice-shelf boundary layer is controlled by vertical mixing and the thermohaline-driven circulation within the sub-ice shelf environment (Rosevear et al., 2024). However, the inflow of ocean water masses into the ice shelf cavities has been used to determine distinct "modes" of basal melting (Jacobs et al., 1992). Mode 1 melting results from the inflow of high-salinity Dense Shelf Water (DSW), formed at winter coastal polynyas. Despite being cold, DSW is above the local freezing point temperatures at depths close to the grounding line. Mode 1 melting is observed within cold ice shelf cavities such as the Filchner-Ronne, Ross and Amery ice shelves (Thompson et al., 2018). Mode 2 melting is driven by the inflow of Circumpolar Deep Water (CDW), which is associated with ice shelves within a warm shelf regime, such as those in the West Antarctic sector (Thompson et al., 2018). Since CDW can be up to 3°C above the local freezing point temperature, Mode 2 is associated with high melt rates, such as found in the Pine Island and Thwaites Ice Shelves in the Amundsen-Bellingshausen sector (Jenkins et al., 2010), and the Mertz and Totten Ice Shelves in East Antarctica (Silvano et al., 2016). Mode 3 melting is associated with shallow melting near the ice shelf calving front, driven by seasonally warmed waters along the coastal currents; this mode has been observed in the ice front of the Filchner-Ronne, Fimbul and Ross ice shelves (Makinson and Nicholls, 1999; Joughin and Padman, 2003; Arzeno et al., 2014; Stewart et al., 2019).

Much of the progress in the understanding of the drivers of basal melting was made through in situ observations. However, these measurements are extremely difficult to obtain and are sparse in both locations and time. Given the harsh conditions, it requires either drilling through hundreds of metres of ice or using autonomous platforms and floats (Miles et al., 2016; Friedrichs et al., 2022; Sallée et al., 2024). Satellite-derived maps of Antarctic basal melting show high spatial variability of the melt rates

(Rignot et al., 2013; Adusumilli et al., 2020), especially between cold and warm ice shelves. However, the temporal variability of these satellite datasets has not been validated yet (Vaňková and Nicholls, 2022). Observational efforts, such as those using the ApRES (autonomous phase-sensitive radar) system (Nicholls et al., 2015) greatly improved the understanding of the intra-annual variability in some ice shelves (Cook et al., 2023; Lindbäck et al., 2019; Davis et al., 2018; Sun et al., 2019), while also highlighting disagreement between satellite and in situ melt rates (Vaňková et al., 2021; Lindbäck et al., 2023). To overcome part of these limitations, here we take advantage of a high-resolution ocean—ice-shelf model to describe the circum-Antarctic basal melting variability at intra-annual scale.

Given the importance of basal melt as a key parameter for Antarctic Ice Sheet stability and its future contribution to global sea level, progress towards a better understanding of the main mechanisms driving heat transport into the ice shelf cavities, the relative importance of different basal melting modes, and physical drivers of the temporal variability of basal melting are urgently needed. This study examines these aspects using a realistic simulation from the Whole Antarctic Ocean Model (WAOM, Richter et al., 2022a), a high-fidelity circum-Antarctica ocean—ice-shelf model of high (4 km) horizontal resolution including tides (Richter et al., 2022b). More details on WAOM and the ocean heat budget analyses used for process-based investigation are presented in Section 2. The annual and seasonal links between ocean warming within the ice shelf cavities and basal melting, and the physical drivers, are presented in Section 3.1. The temporal variability of heat transport processes across different regions is presented in Section 3.2 and their driving mechanisms are explored in Section 3.3, followed by discussion and conclusions in Section 4.

#### 75 2 Methods

The analyses presented here were performed using a simulation from the WAOM setup of the Regional Ocean Modeling System (ROMS Shchepetkin and McWilliams, 2005), which has a grid with a south polar projection of uniform horizontal resolution. WAOM includes ice-shelf thermodynamic interactions via three-equation parameterisation (Galton-Fenzi et al., 2012; Holland and Jenkins, 1999) with static ice shelves from Bedmap2 (Dinniman et al., 2007; Fretwell et al., 2013). Here we use WAOM at 4 km, as the mechanisms described here hold both for coarser (10 km) and finer (2 km) resolutions, which is an intermediate commitment in terms of storage (especially regarding the daily heat budget diagnostics, see below). The WAOM simulation at 10 km resolution was initialised using hydrographic data from the ECCO2 reanalysis (Dee et al., 2011) and spun up for 10 years under a repeat year forcing (RYF), with 2007 selected as a representative year of the present-day state (Richter et al., 2022a). The 4 km resolution WAOM simulation was then initialised from the year 10 of the 10 km run and further spun up for another 12 years using the same RYF. The results presented here are averaged over the final three years of the 4 km simulation (years 10 to 12). A comparison across these three years shows virtually no differences (not shown), indicating that the model had reached a quasi-steady state by the time of analysis.

The ocean surface is forced with prescribed surface heat and salt fluxes associated with sea ice growth and melting derived from sea-ice estimates from satellites (Tamura et al., 2011) — representing important aspects of the ocean-sea ice interaction such as the spatial extent of realistic coastal polynyas, and sites of DSW formation, that are not recoverable using existing sea ice models (Dias et al., 2023). Tidal forcing is included using both velocities and surface elevation at the northern boundaries using 13 major constituents from TPXO7.2 (Egbert and Erofeeva, 2002). Lateral conditions are imposed with daily reanalysis from ECCO2 (Dee et al., 2011), daily surface winds at 10 m are obtained from ERA-Interim — combined with monthly relaxation to SOSE sea surface salinity (Mazloff et al., 2010). The WAOM setup here follows Dias et al. (2023), which differs from Richter et al. (2022a) in the surface fluxes treatment. These differences include the removal of sea surface temperature restoring and reduction of the surface downward (positive) surface heat flux (during summer) to 25% of the original value, helping to decrease surface warm biases. A surface warm bias remains to some extent, likely stemming from the reanalysis forcing (Jacob et al., 2025) and the absence of ocean-atmosphere-sea ice coupling in WAOM. Nevertheless, ERA-Interim and its successor, ERA5, are widely regarded as among the most reliable sources of air-sea flux estimates in the Antarctic margins (Bromwich et al., 2011; Jones et al., 2016). As such, uncertainties in surface fluxes remain a key contributor to the overall uncertainty in our results.

Previous comparison has shown WAOM produces cold bottom temperature biases in the Bellingshausen Sea and, to a lesser extent, in the Amundsen Sea (thus underestimating basal melting in these sectors) and fresher biases in the Weddell Sea (Dias et al., 2023), possibly associated with the surface buoyancy forcing from Tamura et al. (2008). Low formation of DSW in the first version of WAOM (Richter et al., 2022a) has been improved in Dias et al. (2023) by fully accounting for the surface salt flux estimates from satellites, which were limited due to a mismatch between forcing and simulated surface fields. More details on the WAOM configuration and evaluation can be found in Richter et al. (2022a, b) and Dias et al. (2023).

#### 110 2.1 Ocean Heat Budget analyses

100

The ocean heat budget analyses were performed for the entire circumpolar Antarctic continental shelf (including the sub-ice shelf cavities). For ease of analysis, the circumpolar domain was divided into bins of  $3^{\circ}$  longitude (latitudinal bins on the east side of the Antarctic Peninsula due to the shape of Antarctica, Figure 1a). We saved daily outputs of heat flux convergences ( ${}^{\circ}$ C s $^{-1}$ ) for all the temperature equation terms:

115 
$$\underbrace{\frac{\partial}{\partial t} (c_p \ \rho \ \theta)}_{\text{NET}} = \underbrace{-\nabla \cdot (c_p \ \rho \ v\theta)}_{\text{ADV}} - \underbrace{\nabla \cdot (c_p \ \rho \ F)}_{\text{DIFF}} + \underbrace{Q_{\text{sfc}}}_{\text{SHF}}$$
(1)

where  $c_p$  is the specific heat capacity of seawater (3989.25 J kg $^{-1}$  °C $^{-1}$ ),  $\rho$  is the potential density (kg m $^{-3}$ ),  $\theta$  is the potential temperature in degrees Celsius, and v is the speed of the ocean, F represents subgridscale processes, and  $Q_{sfc}$  is the net surface heat flux—which accounts for ocean exchanges with the atmosphere/sea-ice (prescribed) and with the ice shelf (via three-equation parameterisation). On the right-hand side, the product  $v\theta$  represents the heat convergence due to the resolved

model advection (hereafter referred to as ADV), while subgridscale processes (F) not resolved by the model's advection scheme are parameterised as diffusion (DIFF). Laterally, DIFF only includes a Laplacian horizontal mixing scheme with a coefficient of  $20 \text{ m}^2 s^{-1}$  that has a diffusive effect on heat in the down-gradient direction. Vertical mixing associated with the K-profile parameterisation used in the model (KPP; Large et al., 1994) is equal to the surface fluxes when vertically integrated — linked with the buoyancy (Tamura et al., 2011) and mechanical (Dee et al., 2011) forcing (outside the ice shelf cavities) — these are all included in the surface flux term  $Q_{sfc}$  (hereafter called SHF). SHF accounts for both the heat fluxes from sea-ice melt and formation (Tamura et al., 2011) outside of the ice shelf cavities and from the ice shelf melting within the cavities. The left-hand side represents the net heat tendency (NET) and is exactly equal to the sum of all the processes on the right-hand side. All the analyses presented here use daily mean diagnostics outputs from the last three simulated years of the 4 km WAOM simulation (years 10 to 12, Figure A2).

#### 3 Results

# 3.1 Physical drivers of basal melting of the Antarctic ice shelves

The ocean-driven melting of the Antarctic ice shelves has a direct relationship with the heat available — as described by the different modes of melting discussed in Jacobs et al. (1992). Maps of annual ocean heat transport (depth-integrated) and basal melt rates are shown in Figure 1a. Ice shelf basal melting integrated horizontally for each 3° bin is shown in Figure 1a. Basal melt rates higher than ~10 Gt/yr are widespread around the Antarctic continent and are located on well-known ice shelves, such as: Getz, Pine Island, Thwaites, Larsen C, and Filchner-Ronne ice shelves in West Antarctica; and Brunt, Fimbul, Amery, West, Shackleton, Totten, Mertz, and Ross ice shelves in East Antarctica. To identify the seasonal variability in the basal melting, summer (red; defined as December to May) and winter (blue; June to November) averages of the basal melting are shown compared with the annual mean (black). A clear seasonal melting is found in some but not all ice shelves — in particular there is high seasonality along the East Antarctic sector. In contrast, the majority of melting in the West Antarctic sector exhibits relatively stable melt rates throughout the year. An exception is the Bellingshausen Sea, where WAOM at 4 km shows seasonal melt due to regional cold biases; the coarser 10 km WAOM reduces this bias, yielding greater and more stable year-round melt. Based on the melting variability between summer and winter averages compared with the annual average, we classify the Antarctic ice shelves into either a seasonal melt regime or a steady melt regime, shown in green/orange colors in Figure 1a: seasonal melting is defined when the seasonal melt rates (either winter or summer averages) are larger than the annual mean ± 20%; otherwise, it is defined as steady melting. Although we classify the Filchner-Ronne Ice Shelf (FRIS) into the steady regime, this is due to our framework requirement of averaging across the 3° longitude ice shelf cavity bins at FRIS, which leads to cancellation between different phases of melting inside the cavity due to the distinct pathways along which warm anomalies are transported within the cavity (Dias et al., 2023; Vaňková and Nicholls, 2022). Observations show that FRIS exhibits two seasonal maxima in the melting variability (Fig. 7 in Vaňková and Nicholls, 2022).

Figure 1. (a) Maps of annual averaged (over the last three simulated years) ocean heat content vertically-integrated (GJ =  $1x10^9$ J, shown everywhere except at the ice shelf cavities), and annual averaged basal melt rates (m/yr). (b) shows the basal melt rates integrated over each longitudinal bin (Gt/yr, 1 Gt =  $1x10^{12}$  kg), shown as annual (black dashed line), summer (December-May, red line) and winter (June-November, blue line) averages. The ocean heat transport (ADV + DIFF) horizontally- and vertically-integrated for each longitudinal bin (TW, 1 TW =  $10^{15}$ W) is shown in (c) only for the ice shelf cavities and (d) for the whole continental shelf — also shown as annual (black), winter (red) and summer (blue) averages. Subregions selected for the timeseries analyses presented in Figures 3, A4, A5, and A6 are shown in shaded grey in (b). The ice shelf melt regimes were classified into seasonal regime (yellow in panel b; when summer or winter melt rates are equal or larger than the annual mean  $\pm 20\%$  but re-binned into  $9^\circ$  longitude intervals), or steady regime (orange, for the remaining cases).

To understand the drivers of the seasonal melting, the depth-integrated total heat transport convergence (ADV plus DIFF in Eqn. 1) was also integrated horizontally for each bin, first only within the ice shelf cavities (Figure 1c), and also over the whole continental shelf (including the ice shelf cavities, Figure 1d). The surface heat flux (SHF), which has a dominant cooling effect, is analysed separately later in the manuscript (Figure 2). The heat transport term integrated meridionally over the continental shelf describes the effect from the cross-slope heat transport plus the zonal convergence at each longitude bin, while the integral over the ice shelf cavities represents the transport across the ice shelf front plus zonal convergence within the ice shelf cavity. This means that the heat transport convergence includes the effect from water masses traveling zonally along the continental shelf before entering the ice shelf cavity, while these water masses are also subjected to air-sea fluxes within the continental shelf.

This impact of the air-sea fluxes on the heat convergence is substantial, given the differences in magnitude of the heat transport convergence in the continental shelf and within the ice shelf cavities (Figure 1c,d and A1b). The correlation between the annual mean heat transport convergence integrated meridionally over the continental shelf and within the ice shelf cavities (black lines in Figure 1c,d) is indeed low ( $r^2 = 0.12$ ). We note this low correlation could be caused by differences in advective timescales not captured in the time-mean, which does not necessarily imply a weak physical relationship. Besides the low correlation, the seasonality also differs between the continental shelf and ice shelf cavities, where the latter has a clear seasonality in the heat convergence within some ice shelves (e.g., East Antarctica, Figure 1c) but not in all of them, while the heat transport convergence over the continental shelf does not exhibit a consistent seasonality.

High basal melt and heat transport convergence within the ice shelf cavities, however, are closely related, with a time-mean correlation (black lines in Figure 1b,c;  $r^2 = 0.90$ ). Moreover, the seasonality in the heat transport aligns with melt regime classification: strong heat transport seasonality into ice shelves is collocated with seasonal melting (e.g., East Antarctica) and insignificant heat transport seasonality occurs for non-seasonal melt ice shelves in the Amundsen Sea. This indicates that heat transport (mostly controlled by ADV, see Figure 2) primarily controls the seasonality in the basal melt. One exception is the Filchner-Ronne Ice Shelf as previously described, which shows seasonality in the heat transport but not in the basal melting (Figure A5). Another exception is the Fimbul Ice Shelf at  $0^{\circ}$ ; given its location between a seasonal regime in the east and steady regime on the west, our framework (possibly due to the coarse  $3^{\circ}$  longitudinal bins) shows consistent seasonality in the heat (advective) transport but relatively steady melt rates. This could indicate a dominance of basal melt induced by currents rather than thermal driving (e.g., Gwyther et al. 2015).

We now look at the full ocean heat budget to understand the relationship between distinct heat transport processes. Individual processes, i.e., advection, horizontal diffusion and surface fluxes, contribute to the net heat tendency (left-hand side in Eqn. 1). The annual mean ocean heat budget results from a balance between winter and summer contributions. We discuss the seasonal ocean heat budget integrated meridionally only within the ice shelf cavities (Figure 2a) and over the continental shelf (Figure 2b), while the annual budget is shown in the Supplemental Material (Figure A1). The net heat tendency (NET) is the

Figure 2. Ocean heat budget horizontally- and vertically-integrated over longitudinal bins (in TW, 1 TW =  $10^{15}$ W) for the whole continental shelf (including ice shelf cavities, upper panel) and only the ice shelf cavities (lower panel). Processes are shown in green (DIFF), blue (SHF), yellow (ADV) and black (NET). Summer (average December to May of the last three simulated years, solid line) and winter (average June to November, dashed line) are shown. Positive (negative) heat flux convergences denotes warming (cooling).

residual between opposite contributions from the total heat transport (ADV+DIFF) and the surface fluxes (SHF), which is observed both in the integration over the continental shelf and within the ice shelf cavities. The total heat transport represents the sum of horizontal advection (ADV), yellow line plus horizontal diffusion (DIFF), green line; both the total transport and its components converge heat (Figures 1c,d and 2a,b). The total heat transport is counter-balanced by cooling from the net surface heat flux (SHF), blue line). While processes have similar effects on both the heat budget integrated over the entire

continental shelf and within the ice shelf cavities only, these effects result from distinct mechanisms, as described below.

Over the continental shelf, the *SHF* represents atmosphere-ocean and ocean-sea ice heat fluxes, which are the main drivers of the seasonal cycle (blue lines in Figure 2b). Atmosphere-ocean heat flux associated with strong atmospheric cooling during winter reduces the continental shelf temperature while sea ice is formed. Stronger cooling is associated with coastal polynyas, where persistent katabatic winds maintain an ice-free region adjacent to the coast (or ice front or fast-ice), sustaining high seaice production and offshore export (Ohshima et al., 2016). Regions with fewer coastal polynyas, such as around the Dronning Maud Land region (around 0° longitude), present much milder wintertime cooling than some sectors in the East (70-150°E) and West (60-110°W) Antarctic (dashed blue line in Figure 2b; Tamura et al. 2011).

The seasonality in the horizontal advection and diffusion over the continental shelf (Figure 2b) are less evident than in the surface fluxes. DIFF represents parameterised lateral diffusive heat fluxes acting down-gradient at strong frontal regions such as the Antarctic Slope Front, with a negligible seasonal cycle in the Antarctic margins. ADV is associated with the ocean circulation, accounting for both cross-shelf (or cross-calving front) heat transport and the zonal heat convergence associated with the slope and coastal currents. The advection term has substantial seasonal variability, although the direction of seasonal change varies spatially.

In the ice shelf cavities (Figure 2a), both advective (ADV) and diffusive (DIFF) heat transport balance the surface heat flux (SHF). SHF here represents the heat extracted from the ocean to melt ice, which is larger where the total heat transport (ADV+DIFF) is also large (Figure 1c). The seasonal variability of the total heat transport is driven mostly by advection, with only a secondary contribution to the seasonality from diffusion (Figure 2a).

Seasonality of the total heat transport and SHF are only large at those ice shelves with a seasonal melt regime (mostly in the East Antarctic, between 0-150°E, Figure 1d), presenting a maximum in summer and minimum in winter. In contrast, for the steady melt regime in the West Antarctic ice shelves (centred at  $150^{\circ}W$ ,  $130^{\circ}W$ ,  $105^{\circ}W$ ), the total heat transport and the surface heat flux have substantially less seasonality. The results from the ocean heat budget indicate that the temporal variability in the advection term (with a secondary contribution from diffusion) is key to understand the basal melt variability.

# 3.2 Sub-seasonal variability of basal melting

205

To analyse the drivers of sub-seasonal variability of basal melting around Antarctica, we integrate the ocean heat budget terms over eleven selected major ice shelves (grey shadings in Figure 1b), which were classified into 1) seasonal melt or 2) steady melt regimes. We closely investigate the Totten Ice Shelf (as an example of the seasonal regime) and the Getz Ice Shelf (as an example of the steady regime) in this and the next section. Other ice shelf regions are presented in the Supplementary Material

Figure 3. Daily climatology (over the last three simulated years) ocean heat budget horizontally- and vertically-integrated over the vicinity of the Totten Ice Shelf (left column) and the Getz Ice Shelf (right column); shown as examples of seasonal and steady melt regions, respectively. (a) and (b) show the heat budget for the whole continental shelf, (c) and (d) show the heat budget for the ice shelf cavities, in TW (1 TW =  $10^{15}$ W). Net heat tendency (NET) is shown in black lines and processes are shown in blue (SHF), green (DIFF), and orange (ADV); positive (negative) heat convergence translates to a warming (cooling) effect within the integrated region. (e) and (f) show the basal melt rates integrated at full-depth (black, horizontally-integrated) and in the upper-300 m of the ice shelf cavity (300 m, blue dashed line) and below 300 m of the ice shelf cavity (300 m, red dashed line), and sea-ice concentration (cyan, averaged).

#### for completeness.

The daily heat budget for the Totten (1) and Getz (2) ice shelf regions are shown in Figure 3. Most processes described here for these two examples generally hold for other subregions classified within the respective regimes, with the exception of Filchner-Ronne Ice Shelf as mentioned previously. The melting regime in the Totten Ice Shelf (Figure 3, left column), has an elevated basal melt rate of up to 100 Gt/yr between late January and May (from day 1 to around day 150, Figure 3e, black line).

This melt is a response to heat convergence (> 2 TW) driven by the ocean circulation (ADV, Figure 3c). The high temporal variability of the advection component (timescale of a few days) suggests that transient processes such as mesoscale eddies, storm-driven effects on coastal circulation, and/or coastal Rossby waves could be the driver of these warming events. Basal melt starts to decline in April ( $\sim$  day 100), which coincides with the abrupt reduction in the ADV warming and rapid SHF cooling over the continental shelf (Figure 3a,c,e). Melt reaches a minimum (

Figure 4. Maps in the Totten Ice Shelf region of (a) annual bottom temperature (over the last three simulated years), (b) summer (January-May) mean bottom temperature anomaly, (c), summer mean surface heat flux, (e) winter (June-November) mean bottom temperature anomaly, and (f) winter mean surface heat flux. The 1500 m isobath is shown in yellow. Temperature-salinity diagrams with daily values along the ice shelf front for summer (d) and winter (g) averages; annual mean values are shown in grey. The basal melt rates are shown as annual mean (a), summer mean (b), summer anomaly (summer mean - annual mean); (c), winter mean (e), and winter anomaly (f).

In the vicinity of the Getz Ice Shelf (Figure 5), different mechanisms dominate the melting variability. Bottom temperature anomalies show some seasonality, but their actual impact on the melt rates is almost negligible. The surface heat flux at the Amundsen Sea Polynya (Stammerjohn et al., 2015) is nearly as strong as the Dalton Polynya, but there is less evidence of deep convection from the bottom temperature anomalies in winter (Figure 5e.f), due to a stronger vertical stratification of the ocean. Evaluation of the hydrography at the ice shelf front near the Central and Siple Troughs (Figure A16d,e) shows the WAOM underestimates mCDW temperatures (e.g., Dundas et al. 2022) given the model cold biases in the Amundsen Sea (Dias et al., 2023). Still, winter bottom temperature anomalies are concentrated along the shelf break and the Dotson trough and show considerable warming. In summer, mild bottom cold anomalies are found within the region. The cross-shelf transect in front of Getz Ice Shelf shows little temperature variability near the ice shelf front (Figure 6c,d). Importantly, these winter and summer anomalies are away from the ice shelf front, and therefore have little impact on the seasonal shallow melting, as seen for the Totten Ice Shelf. The seasonal variability of the surface heat flux and sea-ice conditions could play a role in the seasonal variations of the shelf circulation via cross-shelf density gradients (Li et al., 2025). Nevertheless, the basal melt maps for both seasons are fairly similar in most regions on the Getz cavity, and in particular for the deep parts of the cavity. Seasonal formation of Antarctic Surface Waters and Winter Waters dominate the upper-150 m, while at depths below 400 m the supply of mCDW is relatively constant through the year (Figure 5d,g). A similar process also dominates other steady regime regions, all located in the Amundsen Sea: the Sulzberger Ice Shelf and the Dotson-Thwaites-Pine Island region (heat budget time series for these regions are shown in Figures A4, A7, A8 and A9).

To conclude, ice shelves within the seasonal melt regime—such as Totten—exhibit a pronounced summer-winter contrast in ocean temperatures near the ice shelf front (Figure 6a,b). During summer, Antarctic Surface Water (AASW) subducts beneath these ice shelves, enhancing shallow melting. In contrast, wintertime coastal polynyas cool the ice front region, effectively shutting down shallow melting (Figure 3e). Conversely, steady-state ice shelves exhibit a weaker seasonal contrast in the upper-300 m ocean temperatures. These ice shelves are consistently influenced by relatively warm waters in the deeper layers near the ice front (Figure 6c,d), supporting sustained deep melting throughout the year (Figure 3f).

#### 4 Discussion and conclusions

This study employed a circum-Antartic ocean–ice-shelf model under a prescribed atmosphere-sea ice repeat year forcing for 2007 in order to investigate the drivers of intra-annual variability of basal melting. Our results show that most Antarctic ice shelves exhibit either a seasonal or steady melt regime over these timescales. The variability of the basal melt is directly driven by the total ocean heat transport within the ice shelf cavities (Figure 1b,c). The melting of ice shelves in East Antarctica is predominantly seasonal, driven by a striking contrast between ice front cooling in winter forced by coastal polynya activity, and summertime warming of waters that penetrate into the upper-300 m portion of the ice shelf cavities driving increased melting — a mechanism described as mode 3 (shallow) melting (Jacobs et al., 1992). In contrast, most of the West Antarctic can be classified as steady melting, mainly controlled by warm deep water inflow via deep troughs that does not have a strong sea-

Figure 5. Maps in the Getz ice shelf region of (a) annual bottom temperature (over the last three simulated years), (b) summer (January-May) mean bottom temperature anomaly, (c), summer mean surface heat flux, (e) winter (June-November) mean bottom temperature anomaly, and (f) winter mean surface heat flux. The 1500 m isobath is shown in yellow. Temperature-salinity diagrams with daily values along the ice shelf front for summer (d) and winter (g) averages; annual mean values are shown in grey. The basal melt rates are shown as annual mean (a), summer mean (b), summer anomaly (summer mean - annual mean); (c), winter mean (e), and winter anomaly (f).

sonality (Walker et al., 2007, 2013). Although these warm intrusions exhibit high-frequency variability, likely associated with wind- and buoyancy-driven shelf circulation (Dotto et al., 2020; Yang et al., 2022), the heat supply is approximately constant, and seasonal variations are less pronounced.

These results indicate a dependency of ice shelf melt variability on the relative strength between melting modes. The seasonal melting regime at the Totten region is dictated by strong shallow melting, where seasonally warmed waters near the ice shelf front control most of the seasonal variability in basal melting. At steady melting regimes, such as in the Getz region, the variability of the CDW-driven melt dominates, and the shallow melting does not present a consistent seasonality. While Dense Shelf Water (DSW) has been associated with mode 1 melt at large ice shelves (Nicholls et al., 2003) such as in the Filchner-Ronne Ice Shelf (Suppl. Material; Figures A5 and A8), our model shows that wintertime convection from coastal polynyas is

**Figure 6.** Transects across the shelf break in the vicinity of the Totten Ice Shelf (a,b) and the Siple Trough at the Getz Ice Shelf (c,d) showing potential temperature averaged over July-August (a,c) and February-March (b,d) for the last three years of the simulation.

important to cool down the ocean near the ice front region and actively interrupt the summertime increases in shallow melting.

We now discuss our circum-Antarctic model results in the context of previous studies. In the Sabrina Coast sector of East Antarctica, the Totten and Moscow University Ice Shelves buttress a marine ice sheet that holds ~3.5 m of global mean sea level. Ground-based measurements show relatively low but highly variable melting of the Totten Ice Shelf (Vaňková et al., 2021). These measurements are in agreement with regional modelling studies (e.g., Gwyther et al., 2014) but at some disagreement with satellite-based estimates that have shown melt peak rates (> 4 m yr<sup>-1</sup>), comparable to those observed in the Amundsen and Bellingshausen Seas (Silvano et al., 2017). Although the Totten or Moscow University region has historically been under-sampled, previous modeling studies have highlighted the role of coastal polynyas—such as the Dalton Polynya—in modulating basal melt variability (Gwyther et al., 2014; Khazendar et al., 2013; Kusahara et al., 2024). More recently, the influence of subglacial meltwater (Gwyther et al., 2023) and intrinsic ocean variability (Gwyther et al., 2018) has also been recognised as potentially important contributors. Additionally, winds may influence at longer, inter-annual timescales (Greene et al., 2017). Coastal polynyas in this study are represented from observational estimates, giving confidence that the model captures, at least partially, the effects from relevant processes, such as the landfast ice effect on coastal polynya locations (Achter

et al., 2022).

There is observational evidence of a seasonal cycle in basal melting in other parts of East Antarctica. In the Maud Land region, mode 3 (shallow) melting was observed beneath the Nivlisen Ice Shelf (Lindbäck et al., 2019), and Sun et al. (2019) reported a similar seasonal pattern under the Roi Baudouin Ice Shelf. In Prydz Bay, seasonality of warm intrusions has been identified from both observations and modelling (Galton-Fenzi et al., 2008; Aoki et al., 2022; Gao et al., 2024). In our model, although deep melting driven by mCDW increases during winter, the overall basal melt seasonality is dominated by enhanced shallow melting in summer. This aligns with observational findings of summer-intensified melting (Aoki et al., 2022; Gao et al., 2024). Mode 3 melting has also been described beneath the Fimbul Ice Shelf in both observations and present-day simulations (Hattermann et al., 2012). Similarly, under the Ross Ice Shelf, high melt variability associated with mode 3 melting has been observed (Stern et al., 2013; Arzeno et al., 2014; Kim et al., 2023), and linked to the influence of the Ross Sea Polynya (Stewart et al., 2019). Overall, most of this observed seasonality across these regions points to increased basal melting during the summer months, consistent with the shallow melting mechanism represented in our model.

In the West Antarctic, the Getz Ice Shelf produces more freshwater than any other ice shelf (Rignot et al., 2013; Jacobs et al., 2013) and influences the regional circulation in the Amundsen and Ross Seas (Nakayama et al., 2014). Accelerated thinning of the Getz has been observed since the 1990s (Paolo et al., 2015) and grounding line retreat since the 2000s (Christie et al., 2018), but lack of bathymetry and ocean measurements results in a poor understanding of the drivers of the Getz basal melting. More recent bathymetric measurements clarified the role of deep troughs under Getz in the CDW inflow to the ice shelf and also subglacial outflow for local melting increases (Wei et al., 2020). In our model, the basal melting at Getz is dominated by CDW-driven melting, with little seasonal variability (Assmann et al., 2019; Jacobs et al., 2013), and that weak polynya activity (and thus shelf cooling) allows for a constant supply of warm water via bottom layers (Silvano et al., 2018).

Several studies demonstrate that the Antarctic Ice Sheet mass loss observed in recent decades in the Amundsen Sea, West Antarctica (Rignot et al., 2013), is driven by warm water inflow causing ice shelf melting and thinning (Walker et al., 2007, 2013; Jacobs et al., 2011). This mechanism responds to large-scale wind variability in the Pacific sector at inter-annual to decadal time scales (Adusumilli et al., 2020; Dotto et al., 2020; Yang et al., 2024) — although the remote drivers are still under debate (Park et al., 2024; Haigh and Holland, 2024). Our model results, forced with a repeat year forcing, suggest that the majority of West Antarctic ice shelves have little seasonal variability (Figure 1d), implying that the melt variability is driven by longer timescales that may also be internally generated (e.g., Gwyther et al., 2018; Holland et al., 2019). Observational evidence of seasonality in the shelf circulation near the Dotson (Yang et al., 2022) and Pine Island ice shelves (Webber et al., 2017) exist and are also found in our model and in previous studies (Kimura et al., 2017). However, the seasonal variability is out-of-phase between these regions and their magnitude are considerably smaller than the melt seasonality (Figure A4, middle panel), resulting in a substantially weaker seasonality than we found for East Antarctic ice shelves.

Although the ice shelves in the Bellingshausen Sea were classified under the seasonal regime in the WAOM 4 km simulation, it is important to note that this region exhibits the most substantial differences across WAOM configurations. At 10 km horizontal resolution, there is substantially higher CDW onshelf transport and basal melt rates compared to the 4 and 2 km simulations (see Dias et al. 2023 and Figure A3). Notably, the 10 km solution aligns more closely with available observations at this region (Schmidtko et al., 2014; Adusumilli et al., 2020), although the Bellingshausen Sea region remains far less observed than the Amundsen Sea (Christie et al., 2016). In particular, the WAOM 10 km simulation shows markedly higher melt rates beneath the George VI Ice Shelf, which is dominated by CDW-driven melt and thus classified under a steady melting regime. This contrasts with the seasonal melting regime classification at 4 km, where cooler shelf waters and reduced CDW inflow prevail (Figure 1b). This discrepancy in melt regime classification across WAOM resolutions highlights an important source of uncertainty in our results, stemming from model resolution and associated biases.

Overall, the results shown in this study suggest an important role of the shallow melting as a source of melt variability at sub-annual timescales. This mode prevails in most of the East Antarctic ice shelves, from the eastern Ross to east of Fimbul and neighboring ice shelves. Our findings also point to the importance of the cooling effect of coastal polynyas to reduce melting near the ice shelf front during winter. In addition, summer warming (Antarctic Surface Waters formation) is concentrated near the icescape (ice shelf front, landfast ice or grounded icebergs), where most cooling takes place in the presence of local coastal polynyas or downstream, given the Antarctic Coastal Current influence (Liu et al., 2024). Given the importance of shallow melt (Jacobs et al., 1992) to basal melt variability, future scenarios where climate change could affect seasonal duration (Mosbeux et al., 2023), summertime warming (Stewart et al., 2019), and coastal polynya activity (Kusahara et al., 2024) can cause large impacts on the melting of the ice shelves and for the buttressing of the Antarctic Ice Sheet. While previous studies indicate a shift between shallow melt to CDW-driven melt associated with future scenarios (Hattermann et al., 2014) and ongoing changes (Lauber et al., 2023), an intensification of shallow melting could enhance an extreme warming scenario (Kusahara et al., 2023), with potential large global sea-level impacts (Edwards et al., 2021; Seroussi et al., 2024).

Common to other modeling studies in such regions with very few observations and large uncertainties, the results described here have some limitations. WAOM has been shown to have cold biases in the West Antarctic and fresh biases in the Weddell Sea (Dias et al., 2023) relative to Schmidtko et al. (2014), both associated with coastal polynya activity in these sectors that are captured in satellite estimates (Tamura et al., 2008), regulating local cooling and affecting CDW on-shelf intrusions. The choice of the repeat year forcing (2007) against a multi-decadal dataset (Schmidtko et al., 2014) is also likely contributing to the aforementioned differences. As coastal polynya location depends on the icescape, including grounded icebergs and fast ice (Cougnon et al., 2017; Achter et al., 2022) that are not captured in climate models (Heuzé, 2021; Dias et al., 2021), the basal melt estimates forced by CMIP5-6 models (Jourdain et al., 2022; Seroussi et al., 2024) will likely under-represent the polynyas' effect on melting near the ice front. Another important aspect is that WAOM does not include coupled sea ice. This can affect air-sea exchanges; for example, WAOM exhibits mild surface warm biases at the Antarctic shelf during summer (December-April) in comparison to the EN4 dataset climatology (not shown). Still, ocean temperatures at the ice shelf front remain colder

than 0.5°C at all regions, consistent with the water mass analyses presented in Section 3.3. WAOM does not restore surface temperature, and the net surface heat flux in summer is dictated by the ERA-Interim reanalysis, which contains uncertainties. In addition to surface buoyancy fluxes, lack of sea ice can also lead to an overestimate of the momentum (wind) effect on ocean currents (Jendersie et al., 2018). Surface wind sensitivity experiments using WAOM demonstrate that this effect can be significant in the West Antarctic sector, but there is less sensitivity in other regions (not shown). This high sensitivity could be contributing to melt variability being less dependent on the surface heat flux, in contrast to the East Antarctica sector. The model's resolution, although relatively fine-scale for circumpolar standards, does not fully resolve eddy processes (Hallberg, 2013), in particular submesoscale eddies near the ice front, which could enhance shallow melting (Friedrichs et al., 2022). The model's vertical coordinates (terrain-following) needs to smooth sharp gradients and can misrepresent the ice shelf front (Naughten et al., 2018; Schnaase and Timmermann, 2019), with potential implications for shallow melting.

This study is the first circum-Antarctic description of the ocean processes driving basal melt at intra-annual timescales, and highlights the importance of shallow melting (Jacobs et al., 1992) to the seasonal variability as a dominant mechanism in the East Antarctic sector. Modelling studies such as undertaken here provide a unique opportunity to describe in detail the basal melting variability and the driving oceanic processes, which is essential to progress our understanding of ocean—ice-shelf interactions given the lack of ocean measurements (Heywood et al., 2014) and bathymetry data along the Antarctic margin (Frémand et al., 2023). Recent studies have also highlighted that circum-Antarctic estimates of basal melting from satellites (e.g., Adusumilli et al., 2020) show large discrepancies from in-situ observations (Vaňková et al., 2021; Lindbäck et al., 2023), again suggesting modelling studies of the ocean processes around Antarctic ice shelves are essential to better understand the drivers of basal melting.

Our results highlight the importance of the seasonal Antarctic melting, modulated by the local icescape (i.e., coastal polynyas activity; Kusahara et al. 2010), to the basal melt variability in a present-day scenario. Climate change-induced impacts on seasonality (Timmermann and Goeller, 2017; Mosbeux et al., 2023) and sea-ice regime shifts (Purich and Doddridge, 2023), in particular anomalous summertime conditions associated with climate and atmospheric circulation anomalies (Clem et al., 2024), can have potentially large implications for shallow melting and thus to Antarctic Ice Sheet mass loss and global sea level rise.

Code and data availability. All the post-processing scripts used in this study are freely available in https://github.com/fabiobdias/waom\_notebook. The model outputs are available from the corresponding author on request.

#### Appendix A: Appendix A

405

**Figure A1.** As in Figure 2 but for the annual mean ocean heat budget horizontally- and vertically-integrated over longitudinal bins (in TW) for the whole continental shelf (including ice shelf cavities, upper panel) and only the ice shelf cavities (lower panel). The heat budget closure is demonstrated by the comparison of the sum of RHS terms of the Eqn. 1 (dotted magenta line) and the NET term.

**Figure A2.** Evolution over the 12-years simulation of the shelf integrated (south of the 1500 m isobath) for (a) sea surface temperature (SST, in  $^{\circ}$ C), (b) ocean heat content (Joules), (c) sea surface salinity (SSS), and (d) ocean salt content.

**Figure A3.** Basal melt rates integrated over each longitudinal bin (Gt/yr, 1  $Gt = 1x10^{12}$  kg) for WAOM10 (10 km horizontal resolution), WAOM4 and WAOM2, shown as annual (black dashed line), January-March (yellow), April-June (blue), July-September (red) and October-December (magenta).

**Figure A4.** As in Figure 3 but showing ocean heat budget (OHB) horizontally- and vertically-integrated over the vicinity of the Sulzberger Ice Shelf (left column), Pine-Island Ice Shelf (middle column) and the Bellingshausen Sea region (right column).

**Figure A5.** As in Figure 3 but showing ocean heat budget (OHB) horizontally- and vertically-integrated over the vicinity of the Filchner-Ronne Ice Shelf (left column), Brunt-Fimbul Ice Shelf (middle column) and Amery-West Ice Shelf region (right column).

**Figure A6.** As in Figure 3 but showing ocean heat budget (OHB) horizontally- and vertically-integrated over the vicinity of the Shackleton Ice Shelf (left column), Mertz Ice Shelf (middle column) and Ross Ice Shelf region (right column).

**Figure A7.** As in Figure 4 but for the Sulzberger ice shelf region.

Author contributions. FBD, AKM, BGF and MHE conceived the study. FBD configured and ran the model simulations, performed the diagnostic analyses, and wrote the paper. All authors discussed the results and reviewed the manuscript draft.

Competing interests. The contact author has declared that none of the authors has any competing interests.

Acknowledgements. This research was undertaken on the National Computational Infrastructure (NCI) in Canberra, Australia, which is supported by the Australian Government. This research was supported by the ARC Australian Centre for Excellence in Antarctic Science (ACEAS; AProject SR200100008), ARC Discovery Project DP190100494 and DP250100759.

Figure A8. As in Figure 4 but for the Pine Island ice shelf region.

# 440 References

Figure A9. As in Figure 4 but for the Bellingshausen Sea region.

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

Figure A10. As in Figure 4 but for the Filchner-Ronne ice shelf region.

- Assmann, K. M., Darelius, E., Wåhlin, A. K., Kim, T. W., and Lee, S. H.: Warm Circumpolar Deep Water at the Western Getz Ice Shelf Front, Antarctica, Geophysical Research Letters, 46, 870–878, https://doi.org/10.1029/2018GL081354, 2019.
  - Bromwich, D. H., Nicolas, J. P., and Monaghan, A. J.: An Assessment of precipitation changes over antarctica and the southern ocean since 1989 in contemporary global reanalyses, Journal of Climate, 24, 4189–4209, https://doi.org/10.1175/2011JCLI4074.1, 2011.
  - Christie, F. D., Bingham, R. G., Gourmelen, N., Tett, S. F., and Muto, A.: Four-decade record of pervasive grounding line retreat along the Bellingshausen margin of West Antarctica, https://doi.org/10.1002/2016GL068972, 2016.
  - Christie, F. D., Bingham, R. G., Gourmelen, N., Steig, E. J., Bisset, R. R., Pritchard, H. D., Snow, K., and Tett, S. F.: Glacier change along West Antarctica's Marie Byrd Land Sector and links to inter-decadal atmosphere-ocean variability, Cryosphere, 12, 2461–2479, https://doi.org/10.5194/tc-12-2461-2018, 2018.
  - Clem, K. R., Raphael, M. N., Adusumilli, S., Amory, C., Baiman, R., Banwell, A. F., Barreira, S., Beadling, R. L., Bozkurt, D., Colwell, S., Coy, L., Datta, R. T., Deb, P., Laat, J. D., du Plessis, M., Fernandez, D., Fogt, R. L., Fricker, H. A., Gille, S. T., Johnson, B., Josey, S. A., Keller, L. M., Kramarova, N. A., Kromer, J., Leslie, R. L., Lazzara, M. A., Lieser, J. L., MacFerrin, M., MacGilchrist, G. M., MacLennan, M. L., Marouchos, A., Massom, R. A., McMahon, C. R., Mikolajczyk, D. E., Mote, T. L., Newman, P. A., Norton, T., Petropavlovskikh, I., Pezzi, L. P., Pitts, M., Reid, P., Santee, M. L., Scambos, T. A., Schulz, C., Shi, J.-R., Souza, E., Stammerjohn,

**Figure A11.** As in Figure 4 but for the Brunt-Fimbul ice shelf region.

- S., Thomalla, S., Tripathy, S. C., Trusel, L. D., Turner, K., and Yin, Z.: Antarctica and the Southern Ocean, Bulletin of the American Meteorological Society, 105, S331–S370, https://doi.org/10.1175/BAMS-D-24-0099.1, 2024.
- Cook, S., Nicholls, K. W., Vaňková, I., Thompson, S. S., and Galton-Fenzi, B. K.: Data initiatives for ocean-driven melt of Antarctic ice shelves, Annals of Glaciology, 45, https://doi.org/10.1017/aog.2023.6, 2023.
- Cougnon, E. A., Galton-Fenzi, B. K., Rintoul, S. R., Legrésy, B., Williams, G. D., Fraser, A. D., and Hunter, J. R.: Regional Changes in Icescape Impact Shelf Circulation and Basal Melting, Geophysical Research Letters, 44, 11,519–11,527, https://doi.org/10.1002/2017GL074943, 2017.
- Davis, P. E., Jenkins, A., Nicholls, K. W., Brennan, P. V., Abrahamsen, E. P., Heywood, K. J., Dutrieux, P., Cho, K. H., and Kim, T. W.: Variability in Basal Melting Beneath Pine Island Ice Shelf on Weekly to Monthly Timescales, Journal of Geophysical Research: Oceans, 123, 8655–8669, https://doi.org/10.1029/2018JC014464, 2018.
- Dee, D. P., Uppala, S. M., Simmons, A. J., Berrisford, P., Poli, P., Kobayashi, S., Andrae, U., Balmaseda, M. A., Balsamo, G., Bauer, P.,
  Bechtold, P., Beljaars, A. C., van de Berg, L., Bidlot, J., Bormann, N., Delsol, C., Dragani, R., Fuentes, M., Geer, A. J., Haimberger,
  L., Healy, S. B., Hersbach, H., Hólm, E. V., Isaksen, L., Kållberg, P., Köhler, M., Matricardi, M., Mcnally, A. P., Monge-Sanz, B. M.,
  Morcrette, J. J., Park, B. K., Peubey, C., de Rosnay, P., Tavolato, C., Thépaut, J. N., and Vitart, F.: The ERA-Interim reanalysis:

**Figure A12.** As in Figure 4 but for the Amery-West ice shelf region.

495

Configuration and performance of the data assimilation system, Quarterly Journal of the Royal Meteorological Society, 137, 553–597, https://doi.org/10.1002/qj.828, 2011.

Dias, F. B., Domingues, C. M., Marsland, S. J., Rintoul, S. R., Uotila, P., Fiedler, R., Mata, M. M., Bindoff, N. L., and Savita, A.: Subpolar Southern Ocean Response to Changes in the Surface Momentum, Heat, and Freshwater Fluxes under 2xCO2, Journal of Climate, 34, 8755–8775, https://doi.org/10.1175/JCLI-D-21-0161.1, 2021.

Dias, F. B., Rintoul, S. R., Richter, O., Galton-Fenzi, B. K., Zika, J. D., Pellichero, V., and Uotila, P.: Sensitivity of simulated water mass transformation on the Antarctic shelf to tides, topography and model resolution, Frontiers in Marine Science, 10, https://doi.org/10.3389/fmars.2023.1027704, 2023.

Dinniman, M. S., Klinck, J. M., and Smith, W. O.: Influence of sea ice cover and icebergs on circulation and water mass formation in a numerical circulation model of the Ross Sea, Antarctica, Journal of Geophysical Research: Oceans, 112, https://doi.org/10.1029/2006JC004036, 2007.

Dotto, T. S., Garabato, A. C. N., Wåhlin, A. K., Bacon, S., Holland, P. R., Kimura, S., Tsamados, M., Herraiz-Borreguero, L., Kalén, O., and Jenkins, A.: Control of the Oceanic Heat Content of the Getz-Dotson Trough, Antarctica, by the Amundsen Sea Low, Journal of Geophysical Research: Oceans, 125, https://doi.org/10.1029/2020JC016113, 2020.

**Figure A13.** As in Figure 4 but for the Shackleton ice shelf region.

Drews, R.: Evolution of ice-shelf channels in Antarctic ice shelves, Cryosphere, 9, 1169–1181, https://doi.org/10.5194/tc-9-1169-2015, 2015.

Dundas, V., Darelius, E., Daae, K., Steiger, N., Nakayama, Y., and Kim, T. W.: Hydrography, circulation, and response to atmospheric forcing in the vicinity of the central Getz Ice Shelf, Amundsen Sea, Antarctica, Ocean Science, 18, 1339–1359, https://doi.org/10.5194/os-18-1339-2022, 2022.

Edwards, T. L., Nowicki, S., Marzeion, B., Hock, R., Goelzer, H., Seroussi, H., Jourdain, N. C., Slater, D. A., Turner, F. E., Smith, C. J., McKenna, C. M., Simon, E., Abe-Ouchi, A., Gregory, J. M., Larour, E., Lipscomb, W. H., Payne, A. J., Shepherd, A., Agosta, C., Alexander, P., Albrecht, T., Anderson, B., Asay-Davis, X., Aschwanden, A., Barthel, A., Bliss, A., Calov, R., Chambers, C., Champollion, N., Choi, Y., Cullather, R., Cuzzone, J., Dumas, C., Felikson, D., Fettweis, X., Fujita, K., Galton-Fenzi, B. K., Gladstone, R., Golledge, N. R., Greve, R., Hattermann, T., Hoffman, M. J., Humbert, A., Huss, M., Huybrechts, P., Immerzeel, W., Kleiner, T., Kraaijenbrink, P., clec'h, S. L., Lee, V., Leguy, G. R., Little, C. M., Lowry, D. P., Malles, J. H., Martin, D. F., Maussion, F., Morlighem, M., O'Neill, J. F., Nias, I., Pattyn, F., Pelle, T., Price, S. F., Quiquet, A., Radić, V., Reese, R., Rounce, D. R., Rückamp, M., Sakai, A., Shafer, C., Schlegel, N. J., Shannon, S., Smith, R. S., Straneo, F., Sun, S., Tarasov, L., Trusel, L. D., Breedam, J. V., van de Wal, R., van den Broeke, M., Winkelmann, R., Zekollari, H., Zhao, C., Zhang, T., and Zwinger, T.: Projected land ice contributions to twenty-first-century sea level rise, Nature, 593, 74–82, https://doi.org/10.1038/s41586-021-03302-y, 2021.

**Figure A14.** As in Figure 4 but for the Mertz ice shelf region.

Egbert, G. D. and Erofeeva, S. Y.: Efficient Inverse Modeling of Barotropic Ocean Tides, Tech. rep., 2002.

Fox-Kemper, B., Hewitt, H., Xiao, C., Aðalgeirsdótti, G., Drijfhout, S., Edwards, T., Golledge, M. H. N., Kopp, R., Krinner, G., Mix, D. N. A., Nowicki, S., Nurhati, I., Ruiz, L., Sallée, J.-B., Slangen, A., and Yu, Y.: Ocean, Cryosphere and Sea Level Change, Tech. rep., https://doi.org/10.1017/9781009157896.011, 2023.

Fraser, A. D., Wongpan, P., Langhorne, P. J., Klekociuk, A. R., Kusahara, K., Lannuzel, D., Massom, R. A., Meiners, K. M., Swadling, K. M., Atwater, D. P., Brett, G. M., Corkill, M., Dalman, L. A., Fiddes, S., Granata, A., Guglielmo, L., Heil, P., Leonard, G. H., Mahoney, A. R., McMinn, A., van der Merwe, P., Weldrick, C. K., and Wienecke, B.: Antarctic Landfast Sea Ice: A Review of Its Physics, Biogeochemistry and Ecology, https://doi.org/10.1029/2022RG000770, 2023.

Fretwell, P., Pritchard, H. D., Vaughan, D. G., Bamber, J. L., Barrand, N. E., Bell, R., Bianchi, C., Bingham, R. G., Blankenship, D. D., Casassa, G., Catania, G., Callens, D., Conway, H., Cook, A. J., Corr, H. F., Damaske, D., Damm, V., Ferraccioli, F., Forsberg, R., Fujita, S., Gim, Y., Gogineni, P., Griggs, J. A., Hindmarsh, R. C., Holmlund, P., Holt, J. W., Jacobel, R. W., Jenkins, A., Jokat, W., Jordan, T., King, E. C., Kohler, J., Krabill, W., Riger-Kusk, M., Langley, K. A., Leitchenkov, G., Leuschen, C., Luyendyk, B. P., Matsuoka, K., Mouginot, J., Nitsche, F. O., Nogi, Y., Nost, O. A., Popov, S. V., Rignot, E., Rippin, D. M., Rivera, A., Roberts, J., Ross, N., Siegert, M. J., Smith, A. M., Steinhage, D., Studinger, M., Sun, B., Tinto, B. K., Welch, B. C., Wilson, D., Young, D. A.,

**Figure A15.** As in Figure 4 but for the Ross ice shelf region.

Xiangbin, C., and Zirizzotti, A.: Bedmap2: Improved ice bed, surface and thickness datasets for Antarctica, Cryosphere, 7, 375–393, https://doi.org/10.5194/tc-7-375-2013, 2013.

Friedrichs, D. M., McInerney, J. B., Oldroyd, H. J., Lee, W. S., Yun, S., Yoon, S. T., Stevens, C. L., Zappa, C. J., Dow, C. F., Mueller, D., Steiner, O. S., and Forrest, A. L.: Observations of submesoscale eddy-driven heat transport at an ice shelf calving front, Communications Earth and Environment, 3, https://doi.org/10.1038/s43247-022-00460-3, 2022.

Frémand, A. C., Fretwell, P., Bodart, J. A., Pritchard, H. D., Aitken, A., Bamber, J. L., Bell, R., Bianchi, C., Bingham, R. G., Blankenship, D. D., Casassa, G., Catania, G., Christianson, K., Conway, H., Corr, H. F. J., Cui, X., Damaske, D., Damm, V., Drews, R., Eagles, G., Eisen, O., Eisermann, H., Ferraccioli, F., Field, E., Forsberg, R., Franke, S., Fujita, S., Gim, Y., Goel, V., Gogineni, S. P., Greenbaum, J., Hills, B., Hindmarsh, R. C. A., Hoffman, A. O., Holmlund, P., Holschuh, N., Holt, J. W., Horlings, A. N., Humbert, A., Jacobel, R. W., Jansen, D., Jenkins, A., Jokat, W., Jordan, T., King, E., Kohler, J., Krabill, W., Gillespie, M. K., Langley, K., Lee, J., Leitchenkov, G., Leuschen, C., Luyendyk, B., MacGregor, J., MacKie, E., Matsuoka, K., Morlighem, M., Mouginot, J., Nitsche, F. O., Nogi, Y., Nost, O. A., Paden, J., Pattyn, F., Popov, S. V., Rignot, E., Rippin, D. M., Rivera, A., Roberts, J., Ross, N., Ruppel, A., Schroeder, D. M., Siegert, M. J., Smith, A. M., Steinhage, D., Studinger, M., Sun, B., Tabacco, I., Tinto, K., Urbini, S., Vaughan, D., Welch, B. C., Wilson, D. S., Young, D. A., and Zirizzotti, A.: Antarctic Bedmap data: Findable, Accessible, Interoperable, and Reusable (FAIR) sharing of 60

**Figure A16.** Temperature-salinity diagrams for the ice front region in the Totten and Getz regions, using the longitudinal bins highlighted in (a). This contrasts with Figures 4 and 5, were the hydrography is obtained using several bins following the grey shades in Figure 1b. (b) January and (c) August climatology of bins 115 and 116, corresponding to the Totten Ice Shelf region. January climatology of the (d) bin 17, corresponding to the Siple Trough, and (e) the bin 18, corresponding to the Central Trough.

- years of ice bed, surface, and thickness data, Earth System Science Data, 15, 2695–2710, https://doi.org/10.5194/essd-15-2695-2023, 2023.
  - Galton-Fenzi, B., Maraldi, C., Coleman, R., and Hunter, J.: The cavity under the Amery Ice Shelf, East Antarctica, Journal of Glaciology, 54, 881–887, https://doi.org/hal-00406802v1, 2008.
- Galton-Fenzi, B. K., Hunter, J. R., Coleman, R., Marsland, S. J., and Warner, R. C.: Modeling the basal melting and marine ice accretion of the Amery Ice Shelf, Journal of Geophysical Research: Oceans, 117, https://doi.org/10.1029/2012JC008214, 2012.
  - Gao, L., Yuan, X., Cai, W., Guo, G., Yu, W., Shi, J., Qiao, F., Wei, Z., and Williams, G. D.: Persistent warm-eddy transport to Antarctic ice shelves driven by enhanced summer westerlies, Nature Communications, 15, https://doi.org/10.1038/s41467-024-45010-x, 2024.
  - Greene, C. A., Blankenship, D. D., Gwyther, D. E., Silvano, A., and van Wijk, E.: Wind causes Totten Ice Shelf melt and acceleration, Science Advances, 3, https://doi.org/10.1126/sciadv.1701681, 2017.
- Gwyther, D. E., Galton-Fenzi, B. K., Hunter, J. R., and Roberts, J. L.: Simulated melt rates for the Totten and Dalton ice shelves, Ocean Science, 10, 267–279, https://doi.org/10.5194/os-10-267-2014, 2014.

- Gwyther, D. E., Galton-Fenzi, B. K., Dinniman, M. S., Roberts, J. L., and Hunter, J. R.: The effect of basal friction on melting and freezing in ice shelf-ocean models, Ocean Modelling, 95, 38–52, https://doi.org/10.1016/j.ocemod.2015.09.004, 2015.
- Gwyther, D. E., O'Kane, T. J., Galton-Fenzi, B. K., Monselesan, D. P., and Greenbaum, J. S.: Intrinsic processes drive variability in basal melting of the Totten Glacier Ice Shelf, Nature Communications, 9, https://doi.org/10.1038/s41467-018-05618-2, 2018.
- Gwyther, D. E., Dow, C. F., Jendersie, S., Gourmelen, N., and Galton-Fenzi, B. K.: Subglacial Freshwater Drainage Increases Simulated Basal Melt of the Totten Ice Shelf, Geophysical Research Letters, 50, https://doi.org/10.1029/2023GL103765, 2023.
- Haigh, M. and Holland, P. R.: Decadal Variability of Ice-Shelf Melting in the Amundsen Sea Driven by Sea-Ice Freshwater Fluxes, Geophysical Research Letters, 51, https://doi.org/10.1029/2024GL108406, 2024.
- Hallberg, R.: Using a resolution function to regulate parameterizations of oceanic mesoscale eddy effects, Ocean Modelling, 72, 92–103, https://doi.org/10.1016/j.ocemod.2013.08.007, 2013.
  - Hattermann, T., Nst, O. A., Lilly, J. M., and Smedsrud, L. H.: Two years of oceanic observations below the Fimbul Ice Shelf, Antarctica, Geophysical Research Letters, 39, https://doi.org/10.1029/2012GL051012, 2012.
  - Hattermann, T., Smedsrud, L. H., Nøst, O. A., Lilly, J. M., and Galton-Fenzi, B. K.: Eddy-resolving simulations of the Fimbul Ice Shelf cavity circulation: Basal melting and exchange with open ocean, Ocean Modelling, 82, 28–44, https://doi.org/10.1016/j.ocemod.2014.07.004, 2014.
    - Hausmann, U., Sallée, J. B., Jourdain, N. C., Mathiot, P., Rousset, C., Madec, G., Deshayes, J., and Hattermann, T.: The Role of Tides in Ocean-Ice Shelf Interactions in the Southwestern Weddell Sea, Journal of Geophysical Research: Oceans, 125, https://doi.org/10.1029/2019JC015847, 2020.
- Heuzé, C.: Antarctic Bottom Water and North Atlantic Deep Water in CMIP6 models, Ocean Science, 17, 59–90, https://doi.org/10.5194/os-17-59-2021, 2021.
  - Heywood, K. J., Schmidtko, S., Heuzé, C., Kaiser, J., Jickells, T. D., Queste, B. Y., Stevens, D. P., Wadley, M., Thompson, A. F., Fielding, S., Guihen, D., Creed, E., Ridley, J. K., and Smith, W.: Ocean processes at the Antarctic continental slope, Philosophical Transactions of the Royal Society A: Mathematical, Physical and Engineering Sciences, 372, https://doi.org/10.1098/rsta.2013.0047, 2014.
- Holland, D. M. and Jenkins, A.: Modeling Thermodynamic Ice–Ocean Interactions at the Base of an Ice Shelf, Journal of Physical Oceanography, 29, 1787–1800, https://doi.org/10.1175/1520-0485(1999)029<1787:MTIOIA>2.0.CO;2, 1999.

- Holland, P. R., Bracegirdle, T. J., Dutrieux, P., Jenkins, A., and Steig, E. J.: West Antarctic ice loss influenced by internal climate variability and anthropogenic forcing, Nature Geoscience, 12, 718–724, https://doi.org/10.1038/s41561-019-0420-9, 2019.
- Jacob, B., Queste, B. Y., and Plessis, M. D. D.: Turbulent heat flux dynamics along the Dotson and Getz ice-shelf fronts (Amundsen Sea, Antarctica), Ocean Science, 21, 359–379, https://doi.org/10.5194/os-21-359-2025, 2025.

- Jacobs, S., Giulivi, C., Dutrieux, P., Rignot, E., Nitsche, F., and Mouginot, J.: Getz Ice Shelf melting response to changes in ocean forcing, Journal of Geophysical Research: Oceans, 118, 4152–4168, https://doi.org/10.1002/jgrc.20298, 2013.
- Jacobs, S. S., Helmer, H. H., Doake, C. S. M., Jenkins, A., and Frolich, R. M.: Melting of ice shelves and the mass balance of Antarctica, Journal of Glaciology, 38, https://www.cambridge.org/core., 1992.
- Jacobs, S. S., Jenkins, A., Giulivi, C. F., and Dutrieux, P.: Stronger ocean circulation and increased melting under Pine Island Glacier ice shelf, Nature Geoscience, 4, 519–523, https://doi.org/10.1038/ngeo1188, 2011.
  - Jendersie, S., Williams, M. J., Langhorne, P. J., and Robertson, R.: The Density-Driven Winter Intensification of the Ross Sea Circulation, Journal of Geophysical Research: Oceans, 123, 7702–7724, https://doi.org/10.1029/2018JC013965, 2018.
  - Jenkins, A., Dutrieux, P., Jacobs, S. S., McPhail, S. D., Perrett, J. R., Webb, A. T., and White, D.: Observations beneath Pine Island Glacier in West-Antarctica and implications for its retreat, Nature Geoscience, 3, 468–472, https://doi.org/10.1038/ngeo890, 2010.
    - Jones, R. W., Renfrew, I. A., Orr, A., Webber, B. G., Holland, D. M., and Lazzara, M. A.: Evaluation of four global reanalysis products using in situ observations in the amundsen sea embayment, antarctica, Journal of Geophysical Research, 121, 6240–6257, https://doi.org/10.1002/2015JD024680, 2016.
  - Joughin, I. and Padman, L.: Melting and freezing beneath Filchner-Ronne Ice Shelf, Antarctica, Geophysical Research Letters, 30, https://doi.org/10.1029/2003GL016941, 2003.
  - Joughin, I., Smith, B. E., Howat, I. M., Scambos, T., and Moon, T.: Greenland flow variability from ice-sheet-wide velocity mapping, Journal of Glaciology, 56, 415–430, https://doi.org/10.3189/002214310792447734, 2010.
  - Jourdain, N. C., Molines, J. M., Sommer, J. L., Mathiot, P., Chanut, J., de Lavergne, C., and Madec, G.: Simulating or prescribing the influence of tides on the Amundsen Sea ice shelves, Ocean Modelling, 133, 44–55, https://doi.org/10.1016/j.ocemod.2018.11.001, 2019.
  - Jourdain, N. C., Mathiot, P., Burgard, C., Caillet, J., and Kittel, C.: Ice Shelf Basal Melt Rates in the Amundsen Sea at the End of the 21st Century, Geophysical Research Letters, 49, https://doi.org/10.1029/2022GL100629, 2022.
  - Khazendar, A., Schodlok, M. P., Fenty, I., Ligtenberg, S. R., Rignot, E., and Broeke, M. R. V. D.: Observed thinning of Totten Glacier is linked to coastal polynya variability, Nature Communications, 4, https://doi.org/10.1038/ncomms3857, 2013.
- Kim, T., Hong, J. S., Jin, E. K., Moon, J. H., Song, S. K., and Lee, W. S.: Spatiotemporal variability in ocean-driven basal melting of cold-water cavity ice shelf in Terra Nova Bay, East Antarctica: roles of tide and cavity geometry, Frontiers in Marine Science, 10, https://doi.org/10.3389/fmars.2023.1249562, 2023.
  - Kimura, S., Jenkins, A., Regan, H., Holland, P. R., Assmann, K. M., Whitt, D. B., Wessem, M. V., van de Berg, W. J., Reijmer, C. H., and Dutrieux, P.: Oceanographic Controls on the Variability of Ice-Shelf Basal Melting and Circulation of Glacial Meltwater in the Amundsen Sea Embayment, Antarctica, Journal of Geophysical Research: Oceans, 122, 10131–10155, https://doi.org/10.1002/2017JC012926, 2017.
  - Kusahara, K., Hasumi, H., and Tamura, T.: Modeling sea ice production and dense shelf water formation in coastal polynyas around East Antarctica, Journal of Geophysical Research: Oceans, 115, https://doi.org/10.1029/2010JC006133, 2010.

- Kusahara, K., Tatebe, H., Hajima, T., Saito, F., and Kawamiya, M.: Antarctic Sea Ice Holds the Fate of Antarctic Ice-Shelf Basal Melting in a Warming Climate, Journal of Climate, 36, 713–743, https://doi.org/10.1175/JCLI-D-22-0079.1, 2023.
  - Kusahara, K., Hirano, D., Fujii, M., Fraser, A. D., Tamura, T., Mizobata, K., Williams, G. D., and Aoki, S.: Modeling seasonal-to-decadal ocean-cryosphere interactions along the Sabrina Coast, East Antarctica, Cryosphere, 18, 43–73, https://doi.org/10.5194/tc-18-43-2024, 2024.
  - Large, W. G., Mcwilliams, J. C., and Doney, S. C.: OCEANIC VERTICAL MIXING: A REVIEW AND A MODEL WITH A NONLO-CAL BOUNDARY LAYER PARAMETERIZATION, Tech. rep., 1994.

- Lauber, J., Hattermann, T., de Steur, L., Darelius, E., Auger, M., Nøst, O. A., and Moholdt, G.: Warming beneath an East Antarctic ice shelf due to increased subpolar westerlies and reduced sea ice, Nature Geoscience, 16, 877–885, https://doi.org/10.1038/s41561-023-01273-5, 2023.
- Li, Z., Wang, C., and Zhou, M.: A Model Analysis of Circumpolar Deep Water Intrusions on the Continental Shelf Break in Amundsen Sea, Antarctica, Journal of Geophysical Research: Oceans, 130, https://doi.org/10.1029/2024JC022210, 2025.
  - Lindbäck, K., Moholdt, G., Nicholls, K. W., Hattermann, T., Pratap, B., Thamban, M., and Matsuoka, K.: Spatial and temporal variations in basal melting at Nivlisen ice shelf, East Antarctica, derived from phase-sensitive radars, Cryosphere, 13, 2579–2595, https://doi.org/10.5194/tc-13-2579-2019, 2019.
  - Lindbäck, K., Darelius, E., Moholdt, G., Vankova, I., Hattermann, T., Lauber, J., and de Steur, L.: Basal melting and oceanic observations beneath Fimbulisen, East Antarctica, https://doi.org/10.22541/essoar.170365303.33631810/v1, 2023.
  - Liu, Y., Nikurashin, M., and Peña-Molino, B.: Seafloor roughness reduces melting of East Antarctic ice shelves, Communications Earth and Environment, 5, https://doi.org/10.1038/s43247-024-01480-x, 2024.
  - Makinson, K. and Nicholls, K. W.: Modeling tidal currents beneath Filchner-Ronne Ice Shelf and on the adjacent continental shelf: Their effect on mixing and transport, Journal of Geophysical Research: Oceans, 104, 13 449–13 465, https://doi.org/10.1029/1999jc900008, 1999.
  - Marsh, O. J., Fricker, H. A., Siegfried, M. R., Christianson, K., Nicholls, K. W., Corr, H. F., and Catania, G.: High basal melting forming a channel at the grounding line of Ross Ice Shelf, Antarctica, Geophysical Research Letters, 43, 250–255, https://doi.org/10.1002/2015GL066612, 2016.
  - Mazloff, M. R., Heimbach, P., and Wunsch, C.: An eddy-permitting Southern Ocean state estimate, Journal of Physical Oceanography, 40, 880–899, https://doi.org/10.1175/2009JPO4236.1, 2010.
  - Merino, N., Sommer, J. L., Durand, G., Jourdain, N. C., Madec, G., Mathiot, P., and Tournadre, J.: Antarctic icebergs melt over the Southern Ocean: Climatology and impact on sea ice, https://doi.org/10.1016/j.ocemod.2016.05.001, 2016.
  - Miles, B. W., Stokes, C. R., and Jamieson, S. S.: Pan-ice-sheet glacier terminus change in East Antarctica reveals sensitivity of Wilkes Land to sea-ice changes, Science Advances, 2, https://doi.org/10.1126/sciadv.1501350, 2016.
- Milillo, P., Rignot, E., Rizzoli, P., Scheuchl, B., Mouginot, J., Bueso-Bello, J., and Prats-Iraola, P.: Heterogeneous retreat and ice melt of Thwaites Glacier, West Antarctica, Science Advances, 5, https://doi.org/10.1126/sciady.aau3433, 2019.
  - Mosbeux, C., Padman, L., Klein, E., Bromirski, P. D., and Fricker, H. A.: Seasonal variability in Antarctic ice shelf velocities forced by sea surface height variations, Cryosphere, 17, 2585–2606, https://doi.org/10.5194/tc-17-2585-2023, 2023.
- Nakayama, Y., Timmermann, R., Rodehacke, C. B., Schröder, M., and Hellmer, H. H.: Modeling the spreading of glacial meltwater from the Amundsen and Bellingshausen Seas, Geophysical Research Letters, 41, 7942–7949, https://doi.org/10.1002/2014GL061600, 2014.

- Naughten, K. A., Meissner, K. J., Galton-Fenzi, B. K., England, M. H., Timmermann, R., Hellmer, H. H., Hattermann, T., and Debernard, J. B.: Intercomparison of Antarctic ice-shelf, ocean, and sea-ice interactions simulated by MetROMS-iceshelf and FESOM 1.4, Geoscientific Model Development, 11, 1257–1292, https://doi.org/10.5194/gmd-11-1257-2018, 2018.
- Nicholls, K. W., Padman, L., Schröder, M., Woodgate, R. A., Jenkins, A., and Øterhus, S.: Water mass modification over the continental shelf north of Ronne Ice Shelf, Antarctica, Journal of Geophysical Research: Oceans, 108, https://doi.org/10.1029/2002jc001713, 2003.

660

670

- Nicholls, K. W., Corr, H. F., Stewart, C. L., Lok, L. B., Brennan, P. V., and Vaughan, D. G.: Instruments and methods: A ground-based radar for measuring vertical strain rates and time-varying basal melt rates in ice sheets and shelves, Journal of Glaciology, 61, 1079–1087, https://doi.org/10.3189/2015JoG15J073, 2015.
- Ohshima, K. I., Nihashi, S., and Iwamoto, K.: Global view of sea-ice production in polynyas and its linkage to dense/bottom water formation, https://doi.org/10.1186/s40562-016-0045-4, 2016.
- Paolo, F. S., Fricker, H. A., and Padman, L.: Volume loss from Antarctic ice shelves is accelerating, Science, 348, 327–331, https://doi.org/10.1126/science.aaa0940, 2015.
- Park, T., Nakayama, Y., and Nam, S. H.: Amundsen Sea circulation controls bottom upwelling and Antarctic Pine Island and Thwaites ice shelf melting, Nature Communications, 15, https://doi.org/10.1038/s41467-024-47084-z, 2024.
- Pritchard, H. D., Ligtenberg, S. R., Fricker, H. A., Vaughan, D. G., Broeke, M. R. V. D., and Padman, L.: Antarctic ice-sheet loss driven by basal melting of ice shelves, Nature, 484, 502–505, https://doi.org/10.1038/nature10968, 2012.
  - Purich, A. and Doddridge, E. W.: Record low Antarctic sea ice coverage indicates a new sea ice state, Communications Earth and Environment, 4, https://doi.org/10.1038/s43247-023-00961-9, 2023.
  - Purich, A. and England, M. H.: Historical and Future Projected Warming of Antarctic Shelf Bottom Water in CMIP6 Models, Geophysical Research Letters, 48, https://doi.org/10.1029/2021GL092752, 2021.
  - Richter, O., Gwyther, D. E., Galton-Fenzi, B. K., and Naughten, K. A.: The Whole Antarctic Ocean Model (WAOM v1.0): Development and evaluation, Geoscientific Model Development, 15, 617–647, https://doi.org/10.5194/gmd-15-617-2022, 2022a.
  - Richter, O., Gwyther, D. E., King, M. A., and Galton-Fenzi, B. K.: The impact of tides on Antarctic ice shelf melting, Cryosphere, 16, 1409–1429, https://doi.org/10.5194/tc-16-1409-2022, 2022b.
- Rignot, E., Jacobs, S., Mouginot, J., and Scheuchl, B.: Ice-Shelf Melting Around Antarctica, Science, 341, 266–270, https://doi.org/10.1126/science.1235798, 2013.
  - Rosevear, M. G., Gayen, B., Vreugdenhil, C. A., and Galton-Fenzi, B. K.: How Does the Ocean Melt Antarctic Ice Shelves?, Annual Review of Marine Science, 14, 37, https://doi.org/10.1146/annurev-marine-040323-074354, 2024.
- Sallée, J.-B., Vignes, L., Minière, A., Steiger, N., Pauthenet, E., Lourenco, A., Speer, K., Lazarevich, P., and Nicholls, K. W.: Subsurface floats in the Filchner Trough provide the first direct under-ice tracks of the circulation on shelf, Ocean Science, 20, 1267–1280, https://doi.org/10.5194/os-20-1267-2024, 2024.
  - Schmidtko, S., Heywood, K. J., Thompson, A. F., and Aoki, S.: Multidecadal warming of Antarctic waters, Science, 346, 1227–1231, https://doi.org/10.1126/science.1256117, 2014.
  - Schnaase, F. and Timmermann, R.: Representation of shallow grounding zones in an ice shelf-ocean model with terrain-following coordinates, Ocean Modelling, 144, https://doi.org/10.1016/j.ocemod.2019.101487, 2019.
    - Seroussi, H., Verjans, V., Nowicki, S., Payne, A. J., Goelzer, H., Lipscomb, W. H., Abe-Ouchi, A., Agosta, C., Albrecht, T., Asay-Davis, X., Barthel, A., Calov, R., Cullather, R., Dumas, C., Galton-Fenzi, B. K., Gladstone, R., Golledge, N. R., Gregory, J. M., Greve, R., Hattermann, T., Hoffman, M. J., Humbert, A., Huybrechts, P., Jourdain, N. C., Kleiner, T., Larour, E., Leguy, G. R., Lowry, D. P.,

- Little, C. M., Morlighem, M., Pattyn, F., Pelle, T., Price, S. F., Quiquet, A., Reese, R., Schlegel, N. J., Shepherd, A., Simon, E., Smith, R. S., Straneo, F., Sun, S., Trusel, L. D., Breedam, J. V., Katwyk, P. V., van de Wal, R. S., Winkelmann, R., Zhao, C., Zhang, T., and Zwinger, T.: Insights into the vulnerability of Antarctic glaciers from the ISMIP6 ice sheet model ensemble and associated uncertainty, Cryosphere, 17, 5197–5217, https://doi.org/10.5194/tc-17-5197-2023, 2023.
- Seroussi, H., Pelle, T., Lipscomb, W. H., Abe-Ouchi, A., Albrecht, T., Alvarez-Solas, J., Asay-Davis, X., Barre, J. B., Berends, C. J., Bernales, J., Blasco, J., Caillet, J., Chandler, D. M., Coulon, V., Cullather, R., Dumas, C., Galton-Fenzi, B. K., Garbe, J., Gillet-Chaulet, F., Gladstone, R., Goelzer, H., Golledge, N., Greve, R., Gudmundsson, G. H., Han, H. K., Hillebrand, T. R., Hoffman, M. J., Huybrechts, P., Jourdain, N. C., Klose, A. K., Langebroek, P. M., Leguy, G. R., Lowry, D. P., Mathiot, P., Montoya, M., Morlighem, M., Nowicki, S., Pattyn, F., Payne, A. J., Quiquet, A., Reese, R., Robinson, A., Saraste, L., Simon, E. G., Sun, S., Twarog, J. P., Trusel, L. D., Urruty, B., Breedam, J. V., van de Wal, R. S., Wang, Y., Zhao, C., and Zwinger, T.: Evolution of the Antarctic Ice Sheet Over the Next Three Centuries From an ISMIP6 Model Ensemble, Earth's Future, 12, https://doi.org/10.1029/2024EF004561, 2024.
- Shchepetkin, A. F. and McWilliams, J. C.: The regional oceanic modeling system (ROMS): A split-explicit, free-surface, topography-following-coordinate oceanic model, Ocean Modelling, 9, 347–404, https://doi.org/10.1016/j.ocemod.2004.08.002, 2005.
  - Silvano, A., Rintoul, S. R., and Herraiz-Borreguero, L.: Ocean-ice shelf interaction in East Antarctica, Oceanography, 29, 130–143, https://doi.org/10.5670/oceanog.2016.105, 2016.
- Silvano, A., Rintoul, S. R., Peña-Molino, B., and Williams, G. D.: Distribution of water masses and meltwater on the continental shelf near the Totten and Moscow University ice shelves, Journal of Geophysical Research: Oceans, 122, 2050–2068, https://doi.org/10.1002/2016JC012115, 2017.
  - Silvano, A., Rintoul, S. R., Peña-Molino, B., Hobbs, W. R., van Wijk, E., Aoki, S., Tamura, T., and Williams, G. D.: Freshening by glacial meltwater enhances melting of ice shelves and reduces formation of Antarctic Bottom Water, Science Advances, 4, https://doi.org/10.1126/sciadv.aap9467, 2018.
- Stammerjohn, S. E., Maksym, T., Massom, R. A., Lowry, K., Arrigo, K. R., Yuan, X., Raphael, M., Randall-Goodwin, E., Sherrell, R. M., and Yager, P. L.: Seasonal sea ice changes in the amundsen sea, Antarctica, over the period of 1979-2014, Elementa, 3, https://doi.org/10.12952/journal.elementa.000055, 2015.
  - Stern, A. A., Dinniman, M. S., Zagorodnov, V., Tyler, S. W., and Holland, D. M.: Intrusion of warm surface water beneath the McMurdo ice shelf, Antarctica, Journal of Geophysical Research: Oceans, 118, 7036–7048, https://doi.org/10.1002/2013JC008842, 2013.
- Stewart, A. L., Klocker, A., and Menemenlis, D.: Circum-Antarctic Shoreward Heat Transport Derived From an Eddy- and Tide-Resolving Simulation, Geophysical Research Letters, 45, 834–845, https://doi.org/10.1002/2017GL075677, 2018.

- Stewart, C. L., Christoffersen, P., Nicholls, K. W., Williams, M. J. M., and Dowdeswell, J. A.: Basal melting of Ross Ice Shelf from solar heat absorption in an ice-front polynya, Nature Geoscience, 12, 435–440, https://doi.org/10.1038/s41561-019-0356-0, 2019.
- Sun, S., Hattermann, T., Pattyn, F., Nicholls, K. W., Drews, R., and Berger, S.: Topographic Shelf Waves Control Seasonal Melting Near Antarctic Ice Shelf Grounding Lines, Geophysical Research Letters, 46, 9824–9832, https://doi.org/10.1029/2019GL083881, 2019.
- Tamura, T., Ohshima, K. I., and Nihashi, S.: Mapping of sea ice production for Antarctic coastal polynyas, Geophysical Research Letters, 35, https://doi.org/10.1029/2007GL032903, 2008.
- Tamura, T., Ohshima, K. I., Nihashi, S., and Hasumi, H.: Estimation of surface heat/salt fluxes associated with sea ice growth/melt in the Southern Ocean, Scientific Online Letters on the Atmosphere, 7, 17–20, https://doi.org/10.2151/sola.2011-005, 2011.
- 725 Thompson, A. F., Stewart, A. L., Spence, P., and Heywood, K. J.: The Antarctic Slope Current in a Changing Climate, https://doi.org/10.1029/2018RG000624, 2018.

- Timmermann, R. and Goeller, S.: Response to Filchner-Ronne Ice Shelf cavity warming in a coupled ocean-ice sheet model Part 1: The ocean perspective, Ocean Science, 13, 765–776, https://doi.org/10.5194/os-13-765-2017, 2017.
- Vaňková, I. and Nicholls, K. W.: Ocean Variability Beneath the Filchner-Ronne Ice Shelf Inferred From Basal Melt Rate Time Series, Journal of Geophysical Research: Oceans, 127, https://doi.org/10.1029/2022JC018879, 2022.

- Vaňková, I., Cook, S., Winberry, J. P., Nicholls, K. W., and Galton-Fenzi, B. K.: Deriving Melt Rates at a Complex Ice Shelf Base Using In Situ Radar: Application to Totten Ice Shelf, Geophysical Research Letters, 48, https://doi.org/10.1029/2021GL092692, 2021.
- Walker, D. P., Brandon, M. A., Jenkins, A., Allen, J. T., Dowdeswell, J. A., and Evans, J.: Oceanic heat transport onto the Amundsen Sea shelf through a submarine glacial trough, Geophysical Research Letters, 34, https://doi.org/10.1029/2006GL028154, 2007.
- Walker, D. P., Jenkins, A., Assmann, K. M., Shoosmith, D. R., and Brandon, M. A.: Oceanographic observations at the shelf break of the Amundsen Sea, Antarctica, Journal of Geophysical Research: Oceans, 118, 2906–2918, https://doi.org/10.1002/jgrc.20212, 2013.
  - Webber, B. G., Heywood, K. J., Stevens, D. P., Dutrieux, P., Abrahamsen, E. P., Jenkins, A., Jacobs, S. S., Ha, H. K., Lee, S. H., and Kim, T. W.: Mechanisms driving variability in the ocean forcing of Pine Island Glacier, Nature Communications, 8, https://doi.org/10.1038/ncomms14507, 2017.
- Wei, W., Blankenship, D. D., Greenbaum, J. S., Gourmelen, N., Dow, C. F., Richter, T. G., Greene, C. A., Young, D. A., Lee, S. H., Kim, T. W., Lee, W. S., and Assmann, K. M.: Getz Ice Shelf melt enhanced by freshwater discharge from beneath the West Antarctic Ice Sheet, Cryosphere, 14, 1399–1408, https://doi.org/10.5194/tc-14-1399-2020, 2020.
  - Yang, H., Kim, T. W., Kim, Y., Yoo, J., Park, J., and Cho, Y. K.: Variability of Inflowing Current Into the Dotson Ice Shelf and Its Cause in the Amundsen Sea, Geophysical Research Letters, 51, https://doi.org/10.1029/2023GL105404, 2024.
- Yang, H. W., Kim, T. W., Dutrieux, P., Wåhlin, A. K., Jenkins, A., Ha, H. K., Kim, C. S., Cho, K. H., Park, T., Lee, S. H., and Cho, Y. K.: Seasonal variability of ocean circulation near the Dotson Ice Shelf, Antarctica, Nature Communications, 13, https://doi.org/10.1038/s41467-022-28751-5, 2022.