# Peer review of "Seasonal variability of ocean heat transport and ice shelf basal melt around Antarctica"

_EGUsphere, 2024_

## Author Comment (AC1)

Response to reviewer comments of egusphere-2024-3905 "On the seasonal variability of ocean heat transport and ice shelf melt around Antarctica" by Fabio Boeira Dias et al.

**REFEREE #1:**

Dear anonymous reviewer,

We thank you for the valuable and constructive feedback. Please find below our responses. In the following, reviewer's comments are shown in **bold font**, our response in regular text, and verbatim changes to the manuscript are indicated with *italic font*.

**General comments**
**In this study the authors produce a pan-Antarctic heat budget using 4km ocean-ice shelf model. They focus on seasonality of this heat transport, with their main results being that ice-shelves in East Antarctica are subject to high seasonality, in contrast to those in West Antarctica. Overall I found this study interesting and easy to follow with clear enough conclusions. While I don't have any particularly major comments, I do have a number of more minor, which I think require addressing before the manuscript be published.**

**Specific comments**
**1. After equation 1, I suggest reminding readers that Q_sfc includes latent heat from imposed sea-ice melt/formation. Further, it was never quite clear to me if this also includes heat fluxes associated with ice-shelf melting. Stating whether or not this is case would be appreciated.**

Thanks for your suggestion. The *Q_sfc* (SHF) includes the latent heat from sea-ice melt and formation and also includes the heat fluxes associated with ice shelf melting. We have included this in the methods (now Lines 124-125):

*"SHF accounts for both the heat fluxes from sea-ice melt and formation (Tamura et al., 2011) outside of the ice shelf cavities and from the ice shelf melting within the cavities."*

**I would suggest including, potentially just as a figure in the appendix, confirmation that the heat budget is closed/accurate. For the ice-shelf cavities, this would require the latent heat flux from the melting of ice shelves. Has this been diagnosed in these simulations (from my comment 1, I wasn't certain it's part of Q_sfc)? Subtracting this from the timeseries in Fig. 1c will lead to a measure of the temporal change in the heat content in the cavities, which can be used as a useful verification of the heat budget itself. If the latent heat flux from ice-shelf melting hasn't been diagnosed, it could be estimated from the ice-shelf melting itself, but this would have small errors since this is also dependent on the local temperature field.**

Thanks for your suggestion. We have included in Figure A1 the annual average heat budget closure, by summing up all RHS terms of Eqn. 1 (DIFF + ADV + SHF, dotted magenta line) and comparing with the

NET term (solid black line). As mentioned in our response to comment #1 above, the net surface heat flux term (SHF) includes the heat flux from ice shelf melting. In agreement with the reviewer suggestion, the black line in Fig. A1 (NET) represents the temporal change in the heat content within the ice shelf cavities (Fig. A1a) and the whole continental shelf (Fig. A1b), and the perfect match between the dotted magenta and solid black lines (in both panels) confirms the heat budget is closed.

**2. L140: "The heat transport term integrated meridionally over the continental shelf describes the effect from the cross-slope heat transport". There's also a contribution coming from heat going into/out of the ice shelf cavities.**

Thanks for your comment, but in this case where we integrated the heat transport over the whole continental shelf (including the ice shelf cavities), the horizontal advective and diffusive component shown in Fig. 1d accounts for the cross-slope and zonal convergence only (but not the cross-calving front), accordingly with the boundaries of each longitudinal bin. The cross-calving front and zonal convergence within adjacent ice shelves (when lacking meridional obstruction) are included in the heat transport term in Fig. 1c.

To further clarify this in the manuscript, we opted to change the labels in Fig. 1d, 2b and A1b from "Continental shelf" to "Continental Shelf (incl. ice shelf)". In addition, we added to Fig. A1b the difference between the integral over the whole continental shelf (including ice shelf cavities) and the integral over the ice shelf cavities only in dashed lines; this is denoted in the legend as, e.g. "NET (excl. ice shelf)". We now mention this figure in lines 161-162:

*"This impact of the air-sea fluxes on the heat convergence is substantial, given the differences in magnitude of the heat transport convergence in the continental shelf and within the ice shelf cavities (Figure 1c,d and A1b)."*

**3. L147. "The correlation between the annual mean heat convergence integrated meridionally over the continental shelf and within the ice shelf cavities is indeed low". Is this the correlation between the black lines in Figs 1c,d? I would imagine advective timescales cause this correlation to be low. I would also imagine that considering lagged correlations wouldn't help much since the lag would be location-dependent. I suggest adding a sentence describing such potential reasons for the low correlation, and how it doesn't necessarily such a weak physical relationship between the timeseries.**

Thank you very much for your suggestion. Yes, you're correct, this correlation refers to the black lines in Fig. 1c,d. We agree with these potential sources of degrading the correlation, and have amended the sentences in lines 163-166:

*"The correlation between the annual mean heat convergence integrated meridionally over the continental shelf and within the ice shelf cavities (black lines in Figure 1c,d) is indeed low ($r^2 = 0.12$). We note that this*

*low correlation could be caused by differences in advective timescales not represented in the time-mean, which does not necessarily imply a weak physical relationship."*

**4. L154,155. Can it be clarified what this "spatial correlation" is referring to? My thinking is that it refers to the correlation between instantaneous maps of heat flux convergence in the ice-shelf cavity and ice-shelf melting, which is then averaged over time. Is this correct?**

Thank you for your comment. This refers to the correlation between the time-mean basal melting (Fig. 1b, black line) and the time-mean heat convergence due to total transport (Fig. 1c, black line). Because we're again looking for the relationship between time-mean variables, we referred to this as spatial correlation. To avoid confusion, we rephrase this sentence (now at lines 171-172), which now reads:

*"High basal melt and heat transport convergence within the ice shelf cavities, however, are closely related, exhibiting a high time-mean correlation (black lines in Figure 1b,c; $r^2$ = 0.90)."*

**5. L196. It's stated that the grey sections in Fig 1b are based on high basal melt rates, but some areas with high basal melt are not included (e.g., near -20), and some grey regions have low basal melt (e.g., 1st and 8th grey section). Can the authors explain the reasoning for this?**

Thank you for your question. We agree with this suggestion that some areas with high basal melt rates are not included. Although we aimed to include all regions with significant basal melt, our choice was focused on major, well-known ice shelves. Thus we have changed the reason for the chosen sub-regions as they are related with major ice shelves (lines 221-222):

*"... we integrate the ocean heat budget terms over eleven selected major ice shelves (grey shadings in Figure 1b)".*

In addition, we have modified the Fimbul sub-region to include the region around longitude -20°, encompassing the Brunt and Riiser-Larsen Ice Shelves. This expanded sub-region was renamed to "Brunt-Fimbul", as the Brunt, Riiser-Larsen and Fimbul Ice Shelves are all classified under a steady melting regime. We also extended the Amery Ice Shelf sub-region (8th grey section) to include the West Ice Shelf; this updated sub-region is now referred to as "Amery-West". Regarding the 1[st] grey region, it corresponds to the eastern side of the Ross Ice Shelf and remains included in the Ross sub-region. It is interconnected with the bin/grey area in the right corner of Fig. 1b. No change has been made to this region.

**6. In section 4 there is some discussion surrounding the use of just one year of model data and associated limitations. I would suggest adding more discussion of how this can also limit confidence in the diagnosed seasonality.**

Thanks for your suggestion. We have expanded the analyses for subsequent years beyond the simulated year 10 shown in the main manuscript. The comparison of the individual annual average ocean heat

budget processes (advection/ADV, diffusion/DIFF and surface heat flux/SHF) is shown in Figure R1, which highlights very minor changes considering the 3 years (after the model spin-up). The advective component (Fig. 2c,d) has slightly larger changes than the diffusive and surface flux terms, but it is still virtually the same across the years analysed. Given that the overall picture is very similar, this indicates that the results analysed in the main manuscript are approximately in a steady state. We have replaced all the figures in the main manuscript and in the Supplemental Material with the averaged over the last three simulated years (years 10 to 12).

We include these results in lines 81-87:

*"The WAOM simulation at 10 km resolution was initialised using hydrographic data from the ECCO2 reanalysis (Dee et al., 2011) and spun up for 10 years under a repeat year forcing (RYF), with 2007 selected as a representative year of the present-day state (Richter et al., 2022a). The 4 km resolution WAOM simulation was then initialised from the year 10 of the 10 km run and further spun up for another 12 years using the same RYF. The results presented here are averaged over the final three years of the 4 km simulation (years 10 to 12). A comparison across these three years shows virtually no differences (not shown), indicating that the model had reached a quasi-steady state by the time of analysis."*

[Figure]

Figure R1: Ocean heat budget (OHB) processes (a,b: diffusion/DIFF; c,d: advection/ADV; e,f: surface flux/SHF) integrated vertically (full-depth) and horizontally over the longitudinal bins for the year 10 (solid lines), 11 (dashed lines) and 12 (dotted lines). OHB horizontally-integrated over the whole continental shelf (a,c,e) and over the ice shelf cavities only (b,d,f). This analysis shows that by Year 10, the model has converged to near steady-state.

**Technical corrections**
**"Antarctic Ice Sheet" is capitalised in places, but not everywhere.**

Thanks, it has now been fixed.

**The phrase "ice-shelf" is hyphenated in instances of "ocean–ice-shelf", but not elsewhere. I suggest sticking with a consistent choice.**

Corrected.

**L18. Change "impede" to "impedes".**

Corrected.

**L23. Change "climate models outputs" to "climate model output".**

Corrected.

**L30. Change ";neither some" to ", nor are some".**

Corrected.

**L51. This paragraph repeats much the previous paragraph, e.g., that warm water ice shelves have mode 2 melting etc. This bit of the text could be made a bit briefer.**

Thanks for your suggestion. We agree with the reviewer's comment, and we have removed the specific description of modes of melting in this paragraph to avoid repetition.

**L79. Specify that these are ice-shelf thermodynamic interactions.**

Thanks, it is now added to the sentence.

**L101. Check the wording of this sentence.**

Thanks, we have amended these sentences, which now reads:

*"The ocean heat budget analyses were performed for the entire circumpolar Antarctic continental shelf (including the sub-ice shelf cavities). For ease of analysis, the circumpolar domain was divided into bins of 3° longitude (latitudinal bins on the east side of the Antarctic Peninsula due to the shape of Antarctica, Figure 1a)."*

**L128. 'Whereas…" This is not a full sentence.**

Thanks for noticing this typo. This sentence now reads:

*"In contrast, the majority of melting in the West Antarctic sector exhibits relatively stable melt rates throughout the year."*

**Fig. 3e,f. Add to the caption the meaning of "<300m" and ">300m" in the legend.**

Corrected, thanks for the suggestion.

**Figs. 4,5. Can either panels b,e (or c,f) be edited to show basal melt anomaly in each season?**

We have modified panels c,f to show the basal melt anomaly from the annual melting, and added this information in the figures' captions (4, 5, A7, A8, A9, A10, A11, A12, A13, A14, A15).

**L280. Typo: "Totten ice Shelf".**

Corrected.

**L289. Typo: "excerce".**

Thanks for catching this. Following also a comment from Reviewer 2, we have amended this passage, which now reads:

*"Although the Totten or Moscow University region has historically been under-sampled, previous modeling studies have highlighted the role of coastal polynyas—such as the Dalton Polynya—in modulating basal melt variability (Gwyther et al., 2014; Khazendar et al., 2013; Kusahara et al., 2024). More recently, the influence of subglacial meltwater (Gwyther et al., 2023) and intrinsic ocean variability (Gwyther et al., 2018) has also been recognised as potentially important contributors. Additionally, winds may influence at longer, inter-annual timescales (Greene et al., 2017)."*

Kind regards,
Fabio Boeira Dias and co-authors

---

## Author Comment (AC3)

Response to reviewer comments of egusphere-2024-3905 "On the seasonal variability of ocean heat transport and ice shelf melt around Antarctica" by Fabio Boeira Dias et al.

**REFEREE #2:**

Dear anonymous reviewer,

Thanks for your comments and constructive feedback. We have addressed your concerns in our responses below, and we believe that clarifying these points has substantially improved the manuscript. In the following, reviewer's comments are shown in **bold font**, our response in regular text, and verbatim changes to the manuscript are indicated with *italic font*.

**Dias et al. present a study of continental shelf and ice shelf cavity ocean heat convergence around Antarctica, highlighting the seasonality of basal melt in East Antarctica in contrast to the more consistent high basal melt rates of West Antarctica. These findings are based on heat budget analysis in a high resolution circum-Antarctic ocean-ice shelf model, WAOM.**

**Overall, the paper is well-written and clear and would make an interesting contribution to TC. However, I do have some concerns about the limitations of the model setup, including the lack of a sea ice model, that I think should be addressed with further discussion on the limitations/implications. I've listed general comments below followed by specific comments related to some interpretations and aspects of the text.**

**General comments:**

**1.  I have some concerns about the model representation that require further model evaluation/description of limitations of the model and how they may affect the results. For example, the T-S diagrams of Getz in Fig. 5 look quite strange and are not consistent with observations (such as in Dundas et al., 2022), so I worry about how representative the model results are. The T-S diagrams for Totten also look strange when compared with Khazendar et al. (2013).**

Thanks for your comment. The T-S diagram of the Getz and Totten regions, shown in Figs. 4 and 5, include a range of longitudinal bins, corresponding to a larger longitudinal range than the observations and model samples mentioned by the reviewer. To offer a fairer comparison with Dundas et al. (2022) and Khazendar et al. (2013) studies, we have reproduced T-S diagrams of the hydrography at the ice shelf front but using a narrow region (Figure R3, following the longitudinal bins highlighted in green in Figure R2).

For the Totten region, we plotted two longitudinal bins (upper row in Figure R3) that encompass the totality of the Totten Ice Shelf front and the eastern region close to the Dalton Polynya. Compared to the modelling results from Khazendar et al. (2013) study (their Fig. 4), the August

climatological mean in WAOM (top right panel, Figure R3 below) shows similar hydrography compared to the green and blue dots (referring to the Dalton Polynya and Totten sub-ice shelf cavity) in Khazendar et al. (2013), their Fig. 4.

For the Getz region, we also plotted T-S diagrams for only two longitudinal bins (bins 17 and 18, corresponding, respectively, to the Siple Trough and Central Trough; lower row in Figure R3) as January climatological mean (annual average also shown in grey dots). These bin locations are in the proximity of the GC6 (Central Trough) and GW6 (Siple Trough) moorings presented in Dundas et al. (2022). The water masses reproduced by WAOM at the Getz Ice Shelf front are modified Circumpolar Deep Water (mCDW), Winter Water (WW) and also Ice Shelf Water (ISW), although with biases in comparison with Dundas et al (2022). In WAOM, mCDW can be up to 1°C colder than in observations, which aligns with the cold bias described in West Antarctica (original manuscript, lines 346-348). This bias compares to biases of the order of 0.5-2°C in CMIP6 models in this region (Purish and England 2021; their Fig. S16). WAOM simulates Ice Shelf Water (temperature colder than -1.9°C), which is not observed in the GC6 and GW6 moorings from Dundas et al. (2022), while its moderately high salinity (around 34.5) indicates some influence from High Salinity Shelf Water, likely sourced from the Amundsen Sea Polynya.

We have added Figures R2 and R3 combined to the Supplemental Material (as Figure A16), and it is now mentioned in the following sentences in the text (lines 265-266 and 279-281):

*"Additional evaluation of the hydrography at the ice shelf front in the Totten is presented in Figure A16b,c—showing a good agreement with previous modeling studies (e.g., Khazendar et al. 2013)."*

*"Evaluation of the hydrography at the ice shelf front near the Central and Siple Troughs (Figure A16d,e) shows the WAOM underestimates mCDW temperatures (e.g., Dundas et al. 2022) given the model cold biases in the Amundsen Sea (Dias et al., 2023)."*

[Figure]

Figure R2: as Fig. 1a in the main manuscript, but highlighting the chosen longitudinal bins in front of the Getz and Totten Ice Shelves, in green boxes.

[Figure]

Figure R3: T-S diagrams using the hydrography at the ice shelf front for the specific longitudinal bins highlighted in green in Fig. R2: (top left) January and (top right) August climatology of bins 115 and 116, corresponding to the Totten Ice Shelf region. January climatology of the (c) bin 17, corresponding to the Siple Trough, and (d) the bin 18, corresponding to the Central Trough.

**2. Interpretation of the surface heat fluxes is complicated by the lack of a sea ice model and the sea surface temperature relaxation applied. I'd appreciate some clarification on:**
- **The impact of the monthly relaxation to SOSE SSTs on the key results**

The original WAOM used in Richter et al (2022a, 2022b) has sea surface temperature (SST) and salinity (SSS) restoring towards SOSE reanalysis. However, in the present study, we used the updated WAOM as described in Dias et al (2023), which does not include SST restoring (only SSS). In the original manuscript, lines 91-93 (copied below), we mistakenly mentioned SST relaxation where we originally meant SSS relaxation. This has now been corrected.

We also added more details on the WAOM setup differences between the current study and Richter et al. (2022a, 2022b), which now reads:

*"Lateral conditions are imposed with daily reanalysis from ECCO2 (Dee et al., 2011), daily surface winds at 10 m are obtained from ERA-Interim — combined with monthly relaxation to SOSE sea surface salinity (Mazloff et al., 2010). The WAOM setup here follows Dias et al. (2023), which differs from Richter et al. (2022a) in the treatment of surface fluxes. These differences include the removal of sea surface temperature restoring, and reduction of the surface downward (positive) surface heat flux (during summer) to 25% of the original value, helping to decrease surface warm biases."*

We expanded the discussions on the WAOM limitations in lines 346-359 of the original manuscript (now at lines 392-405):

Before:
*"WAOM has been shown to have cold biases in the West Antarctic and fresh biases in the Weddell Sea (Dias et al., 2023) relative to Schmidtko et al. (2014), both associated coastal polynya activity in these sectors that are captured in satellite estimates (Tamura et al., 2008) and contributes to local cooling and less CDW on-shelf intrusions.*
*…*
*Another important aspect is that WAOM does not include coupled sea-ice and so can overestimate the momentum (wind) effect on ocean currents (Jendersie et al., 2018). Surface wind sensitivity experiments using WAOM show this effect can be important in the West Antarctic sector, but there is less sensitivity in other regions (not shown)."*

Now reads:
*"WAOM has been shown to have cold biases in the West Antarctic and fresh biases in the Weddell Sea (Dias et al., 2023) relative to Schmidtko et al. (2014), both associated with coastal polynya activity in these sectors that are captured in satellite estimates (Tamura et al., 2008), regulating local cooling and affecting CDW on-shelf intrusions.*
*…*
*Another important aspect is that WAOM does not include coupled sea ice. This can affect air-sea exchanges; for example, WAOM exhibits mild surface warm biases at the Antarctic shelf during summer (December-April) in comparison to the EN4 dataset climatology (not shown). Still, ocean temperatures at the ice shelf front remain colder than 0.5°C at all regions, consistent with the water mass analyses presented in Section 3.3. WAOM does not restore surface temperature, and the net surface heat flux in summer is dictated by the ERA-Interim reanalysis, which contains uncertainties. In addition to surface buoyancy fluxes, lack of sea ice can also lead to an overestimate of the momentum (wind) effect on ocean currents (Jendersie et al., 2018). Surface wind sensitivity experiments using WAOM demonstrate that this effect can be significant in the West Antarctic sector, but there is less sensitivity in other regions (not shown)."*

**- How realistic are the surface heat flux estimates given that the model runs with sea surface temperature correction/restoring? How do you distinguish the regions with strong seasonality from the regions that just require strong sea surface temperature restoring (possibly due to model drift)?**

As mentioned in the previous comment, the WAOM setup used in this study does not include SST relaxation; thus, there is no regional impact from sea surface temperature restoring.

**- Looking at the heat budget figures for each of the different ice shelf regions in the appendix, the main signal is essentially whether or not there is sea ice cover and I'm wondering to what extent that signal is present due to the way sea ice is treated in this model (mentioned a bit in line 218-222)**

Thanks for your comment. We agree that sea ice and its coverage have an important role in the seasonality of basal melting, as the reviewer suggests, which is also shown in the timeseries (Figures 3, A4-A6) and is mentioned in lines 216-221 of the original manuscript. The way sea ice is prescribed in WAOM can thus impact the basal melting. In our model, the sea ice fluxes are prescribed in the surface forcing: i.e. Tamura et al. (2008, 2011) and ERA-Interim reanalysis. The Tamura et al. dataset directly accounts for satellite estimates of sea-ice concentration. ERA-Interim reanalysis indirectly accounts for sea-ice observations via assimilation of the Operational Sea Surface Temperature and Sea Ice Analysis (OSTIA) product. We acknowledge that our setup has its limitations, and a dedicated paragraph discussing the model limitations is included in the manuscript; the lines 395-406 are particularly relevant here:

*"As coastal polynya location depends on the icescape, including grounded icebergs and fast ice (Cougnon et al., 2017; Achter et al., 2022) that are not captured in climate models (Heuzé, 2021; Dias et al., 2021), the basal melt estimates forced by CMIP5-6 models (Jourdain et al., 2022; Seroussi et al., 2024) will likely under-represent the polynyas' effect on melting near the ice front. Another important aspect is that WAOM does not include coupled sea ice. This can affect air-sea exchanges; for example, WAOM exhibits mild surface warm biases at the Antarctic shelf during summer (December-April) in comparison to the EN4 dataset climatology (not shown). Still, ocean temperatures at the ice shelf front remain colder than 0.5°C at all regions, consistent with the water mass analyses presented in Section 3.3. WAOM does not restore surface temperature, and the net surface heat flux in summer is dictated by the ERA-Interim reanalysis, which contains uncertainties. In addition to surface buoyancy fluxes, lack of sea ice can also lead to an overestimate of the momentum (wind) effect on ocean currents (Jendersie et al., 2018). Surface wind sensitivity experiments using WAOM demonstrate that this effect can be significant in the West Antarctic sector, but there is less sensitivity in other regions (not shown). This high sensitivity could be contributing to melt variability being less dependent on the surface heat flux, in contrast to the East Antarctica sector."*

Nevertheless, given that this setup accounts for sea-ice observations, its location and timing are expected to be more realistic when compared to dynamical models, where sea ice often shows large biases which can potentially affect the seasonality driven by sea-ice coverage. In addition, as we are investigating the steady state and not perturbed simulations, the ocean response coupled to the sea ice can be considered less important.

**3.   The link suggested between surface fluxes and bottom temperature changes (cooling effect of polynyas) is not very clearly demonstrated by the figures. Similarly, the links between surface fluxes and inferring what happens in the cavities mechanistically isn't very solid. I think some further analysis or maybe clearer demonstration would help support this point.**

Thanks for the feedback. In the presence of coastal polynyas, we expect a clear link between negative surface heat fluxes and cold waters throughout the water column, resulting from the convective process. To present further evidence of this mechanism, we show below some transects in the vicinity of the Totten and the Getz Ice Shelves (Figure R4).

In the Totten region during winter (July-August climatology, Fig. R4a), the transect shows cold waters (below -1.7°C) in abundance in the entire water column in the area close to the ice shelf front. This transect is located just west of the Dalton Polynya (Fig. 4f in the main manuscript). During summer (February-March, Fig. R4b), surface warming occurs in the upper 50 m; temperatures at the depth of the ice shelf front (around 250-300m in the Totten Ice Shelf) are substantially warmer than in winter, between -0.5°C and -1°C, which is sufficient to induce shallow melting.

On the other hand, the transect near the Getz Ice Shelf shows less evidence of seasonality. The transect crosses the Siple Trough (Fig. R4c,d) and shows relatively warm waters infiltrating the ice shelf cavity during both summer (February-March) and winter (July-August) months. Summertime warming in the upper-50 m also occurs in the Siple Trough transect, but it is less pronounced than in the Totten region, resulting in less variability in the shallow melting (see Fig. 3f in the main manuscript).

These results reinforce that, in East Antarctica (e.g., Totten), the shallow melting is strongly modulated by summertime warming and wintertime cooling, with coastal polynyas playing a major role in shutting down shallow melting. In the Amundsen Sea (e.g., Getz), the melting variability is more stable in our Repeat Year Forcing simulations, as deep warm water more constantly reaches the Getz Ice Shelf and sustains steady melt rates. In contrast, the summertime warming is less pronounced than in East Antarctica.

[Figure]

Figure R4: Transect across the shelf break in the vicinity of the Totten Ice Shelf (upper row) and the Siple Trough in the Getz Ice Shelf (lower row) showing potential temperature averaged over July-August (left column) and February-March (right column).

These transects have been added to the main manuscript as Figure 6, and the following passages were also added:

*"A cross-shelf transect in front of the Totten (Figure 6a) shows further evidence of winter convection near the ice shelf front."* (lines 254-255)

*"The cross-shelf transect in front of Getz Ice Shelf shows little temperature variability near the ice shelf front (Figure 6c,d)"* (lines 282-283)

*"To conclude, ice shelves within the seasonal melt regime—such as Totten—exhibit a pronounced summer-winter contrast in ocean temperatures near the ice shelf front (Figure 6a,b). During summer, Antarctic Surface Water (AASW) subducts beneath these ice shelves, enhancing shallow melting. In contrast, wintertime coastal polynyas cool the ice front region, effectively shutting down shallow melting (Figure 3e). Conversely, steady-state ice shelves exhibit a weaker seasonal contrast in upper-300 m ocean temperatures. These ice shelves are consistently influenced by relatively warm waters in the deeper layers near the ice front (Figure 6c,d), supporting sustained deep melting throughout the year (Figure 3f)."* (lines 293-298)

**4. The splitting of the heat fluxes into which components dominate is an important portion of the study, but is occasionally challenging to follow because of all the acronyms and nuances to the interpretation due to the model setup. For example, in line 160-175 I find the**

**descriptions of the different components a bit difficult to follow. I'm not sure what the best solution is, but you could consider adding either another equation that shows each of these components (or annotating Eqn. 1), or a table/diagram as a visual aid to follow the heat budget components.**

Thanks for your suggestions. We have annotated the Equation 1 with the acronyms for the heat flux processes:

$$\underbrace{\frac{\partial}{\partial t} \left( c_p \, \rho \, \theta \right)}_{\text{NET}} = \underbrace{- \nabla \cdot \left( c_p \, \rho \, \boldsymbol{v} \theta \right)}_{\text{ADV}} - \underbrace{\nabla \cdot \left( c_p \, \rho \, \boldsymbol{F} \right)}_{\text{DIFF}} + \underbrace{Q_{\text{sfc}}}_{\text{SHF}}$$

We have also refined the description of these components to clarify the role of individual processes at both the continental shelf and in the ice shelf cavities, which now reads (lines 186-196):

*"The net heat tendency (NET) is the residual between opposite contributions from the total heat transport (ADV + DIFF) and the surface fluxes (SHF), which is observed both in the integration over the continental shelf and within the ice shelf cavities. The total heat transport represents the sum of horizontal advection (ADV, yellow line) plus horizontal diffusion (DIFF, green line); both the total transport and its components converge heat (Figures 1c,d and 2a,b). The total heat transport is counter-balanced by cooling from the net surface heat flux (SHF, blue line). While processes have similar effects on both the heat budget integrated over the entire continental shelf and within the ice shelf cavities only, these effects result from distinct mechanisms, as described below.*

*Over the continental shelf, the SHF represents atmosphere-ocean and ocean-sea ice heat fluxes, which are the main drivers of the seasonal cycle (blue lines in Figure 2b). Atmosphere-ocean heat flux associated with strong atmospheric cooling during winter reduces the continental shelf temperature while sea ice is formed."*

**Specific comments:**

**Since the focus of the paper is on basal melt, I'd suggest specifying basal melt in the title.**

Thanks for the suggestion. We changed the title to "Seasonal variability of ocean heat transport and ice shelf basal melt around Antarctica".

**Introduction section:**

**Line 33: "A leading source of the uncertainty is due to the differences between ice sheet models" Please include specifics for the differences between ice sheet models you are referring to.**

We have amended this sentence to expand on the sources of uncertainties in ISMIP6 models, which now reads:

*"The leading sources of the uncertainty result from the choice of ice sheet models, followed by uncertainties in ocean-ice interactions and the choice of climate model (Seroussi et al., 2023), pointing to the critical role of limitations in the ocean forcing."*

**Overall, the introduction is strong (especially the second paragraph!) but it would benefit from some links for the 3rd and 4th paragraphs with the rest as they sort of stand alone.**

Thanks for the feedback. We have improved the 3rd and 4th paragraphs aimed at better connecting with the remainder of the introduction.

**Methods section:**

**Line 82-85: Consider rephrasing or re-ordering these sentences as it took a couple of re-reads to get the method for initialisation and that there is no interannual variation in forcing.**

We re-wrote these sentences to clarify the model design and spin-up.

**Line 93: I suggest splitting the paragraph here to create a separate paragraph focussed on the key points of the model evaluation that you highlighted and expanding on it to provide further context for your findings (for example to address the general comment about water mass representation).**

We have split the paragraph and expanded on the model biases. We included the following sentences that are particularly relevant to the main findings:

*"A surface warm bias remains to some extent, likely stemming from the reanalysis forcing (Jacobs et al. 2025) and the absence of ocean-atmosphere-sea ice coupling in WAOM. Nevertheless, ERA5 and its successor, ERA-Interim, are widely regarded as among the most reliable sources of air-sea flux estimates in the Antarctic margins (Bromwich et al. 2011; Jones et al. 2016)."*

**Line 105: Heat convergence equation. Should this also include a term of artificial heat addition/removal due to the monthly relaxation to SOSE surface temperatures or is that already included in the net surface heat flux term? If it is included in the surface heat flux term, it would be good to clarify that here.**

As clarified previously, the model setup does not include SST relaxation (in contrast with the setup in Richter et al. 2022a). Therefore, the only terms included in the net surface heat fluxes are ocean-atmospheric and ocean-sea ice fluxes (calculated by bulk formula in response to the ERA-Interim

and Tamura et al. 2011 forcing), and ocean-ice shelf fluxes (provided by the three-equation parameterisation). We included a few more details in Section 2.1 (lines 117-119 and 122-126 of the revised manuscript).

**Line 115:**
**Why is only the last year of the simulations used instead of the monthly mean climatology over the full set? Are the earlier years still responding to the change in conditions, so effectively spin up? If that is the case, what evidence are you using to suggest that the results are no longer dependent on the initial state at the end of spin up?**

Thanks for your question. We avoid analysing the first years of the model simulation at 4 km resolution as they are effectively spin-up years. We also avoid saving daily budget diagnostics during the full period to save storage resources. We agree that monthly mean diagnostics over a longer period than a year could be useful, but years up until Year 10 were not saved. Instead, we have added more analyses comparing the results from year 10 (used in the main manuscript) with subsequent years (11 and 12) in Figure R1 of this document, where we addressed the similar question from Reviewer #1. Given the similarities in the ocean heat budgets among years 10, 11 and 12, this suggests that the model is effectively close to a steady state during the year analysed (Yr 10). We have replaced the figures in the main manuscript and supplemental material now using the average over the last 3 years (year 10 to 12), and updated the description in the methods section:

**An additional thought, if the 10 year period is not spin up: an improvement to indicate uncertainty in the heat fluxes would be to edit the figures to include solid lines based on the model mean and shading to indicate the range of basal melt rates / heat convergences over the multiple years of running the model.**

We appreciate the suggestion. However, given that we did not save monthly diagnostics during the spin-up years, and that these would incur substantial computational resources, we refer to our response to the question above, where we show that the analyses present in the original manuscript were made under steady-state conditions.

**Results section**

**Line 127: "A clear seasonally cycle is found in some but not all." Suggest specifying that you are referring to a seasonal cycle in basal melt.**

Thanks, we modified this sentence, which now reads:

*"A clear seasonal melting is found in some but not all ice shelves…"*

**Line 128-137: The Bellingshausen Sea is a notable exception that requires clear comment here (mentioned later on), otherwise it seems like the classification is fitted to match the east-west divide rather than coming directly out of the findings**

Thanks for your comment. We agree that a mention of the exception of the Bellingshausen Sea is worth mentioning earlier in the manuscript. We have added some words on these at lines 140-142:

*"An exception is the Bellingshausen Sea, where WAOM at 4~km shows seasonal melt due to regional cold biases; the coarser 10~km WAOM reduces this bias, yielding greater and more stable year-round melt."*

**Figure 1: Great figure overall and I really appreciate the region definitions shown here! The caption says "green in panel b", maybe this is my monitor, but I only see orange and yellow for the steady and seasonal regime; consider using a different color?**

The green colour chosen previously for the seasonal regime was actually "greenyellow" from matplotlib. To avoid confusion, we changed this to "yellow" and also replaced this in the text and figure captions.

**Line 142: Possibly I'm misunderstanding something, but based on the heat convergence calculation, I don't think you can distinguish offshore cooling from cavity melt related cooling. Is that true? If so, it would be good to add a comment here to help interpretation.**

Thanks for your question. The total heat transport mentioned in this paragraph and Figure 1c,d accounts for ADV and DIFF terms in Eqn. 1. The cavity melt-related cooling is accounted in the SHF term of Eqn. 1 (shown in Figure 2 and A1, blue lines); and the surface cooling at the continental shelf (i.e. outside ice shelf cavities) can be estimated by the difference between surface cooling at the whole continental shelf and the surface cooling within the ice shelf cavities (Fig. A1B, solid and dashed blue lines). As this term is analysed later in the manuscript (in the description of Fig. 2), we add the following sentence in this paragraph:

*"The surface heat flux SHF, which has a dominant cooling effect, is analysed separately later in the manuscript (Figure 2)."*

**Line 153: "heat convergence over the continental shelf does not exhibit a clear seasonality" Suggest replacing clear with consistent, since in some places there does seem to be a strong seasonal difference.**

Thanks for the suggestion, we have replaced "clear" with "consistent".

**Fig. 2: Panel (b) shows the strongest magnitude of seasonal difference between summer and winter on the continental shelf in West Antarctica, whereas the NET seasonal difference is**

**smaller and varies less in East Antarctica. I think this requires some further interpretation/discussion in the text as it could contradict the argument of strongest seasonality in East Antarctica that you are presenting.**

Thanks for raising this concern. Firstly, we want to remind the reviewer that the NET term is the residual between warming from the heat transport (i.e. both advective/ADV and diffusive/DIFF components) and cooling from the surface heat fluxes (SHF). The NET term is not driving the melt variability, which is demonstrated by the role of the heat transport in driving basal melt as shown in Figure 1b,c and described in lines 170-174:

*"High basal melt and heat transport convergence within the ice shelf cavities, however, are closely related, with a time-mean correlation (black lines in Figure 1b,c; r2 = 0.90). Moreover, the seasonality in the heat transport aligns with melt regime classification: strong heat transport seasonality into ice shelves is collocated with seasonal melting (e.g., East Antarctica) and insignificant heat transport seasonality occurs for non-seasonal melt ice shelves in the Amundsen Sea. This indicates that heat transport (mostly controlled by ADV, see Figure 2) primarily controls the seasonality in the basal melt."*

The melt effect (i.e., beneath the ice shelves) is represented by the SHF. We can theoretically have a large change in melt (SHF on Fig. 2a) without any change in NET, i.e. no change in temperature, when the heat transport is also large, which in our analyses is often the case. For this reason, we do not agree that the large seasonal variability in the NET contradicts our argument of the strongest basal melting seasonality in East Antarctica. The large seasonality in East Antarctica is clearly demonstrated by the net surface heat flux terms (SHF term within the ice shelf cavities, Fig. 2a) and the heat transport (ADV plus DIFF, Fig. 2a).

We also want to offer a reminder on the role of the surface fluxes (SHF) at the continental shelf (Fig. 2b) versus within the ice shelves (Fig. 2a). The seasonality in SHF over the continental shelf is much larger (about 1-2 TW) than in the ice shelf cavities (~0.5 TW), which results from the different processes represented by SHF in these distinct regions. Over the whole continental shelf, the SHF is dominated by the (prescribed) air-sea-sea ice fluxes, mostly associated with cooling at coastal polynyas. Within the ice shelf cavities, SHF only represents the cooling from basal melting. The large seasonality noticed in the NET term by the reviewer reflects the seasonality in the SHF at the continental shelf, related to the cooling at coastal polynyas (stronger in winter and weaker in summer). However, this does not necessarily imply large seasonality in the basal melting, which is shown to be primarily controlled by the heat transport within the ice shelves.

**Can you comment a bit more about what's going on around Fimbul (0 longitude) in the ice shelf cavity transport in Fig. 2?**

Thanks for your question. Fimbul Ice Shelf and the Brunt-Fimbul region (selected using the longitudinal bin framework) are classified under the seasonal regime on its very eastern side and

under the steady regime on the west/center side (Fig. 1b). From west to east in the Brunt-Fimbul region (i.e., 26°W to 2°E), there is an increase in the seasonality of the advective transport (Fig. 2a). In the west side, closer to the Brunt Ice Shelf, the advective heat transport exhibits little seasonality. On the east side (close to 0°), *ADV* enhances during summer and weakens during winter, as expected for the seasonal melt regime. In addition to the substantial shallow melting during summer, which occurs widespread in East Antarctica (as extensively described in the manuscript), this region also experiences a substantial inflow of modified Warm Deep Water (mWDW) in the deep parts of the Fimbul Ice Shelf (e.g., Hattermann et al. 2012). The inflow of mWDW helps to sustain relatively high melt rates at larger depths of the Fimbul Ice Shelf cavity all year long (Fig. A5h, red dashed line), especially in the main sill where mWDW inflow follows deep bathymetric contours (Fig. A11b,e). We acknowledge that Fimbul is located at the limit between the seasonal regime, found in East Antarctica, and the steady regime in West Antarctica, which can be confusing. We note this characteristic of the Fimbul Ice Shelf at lines 176-179:

*"Another exception is the Fimbul Ice Shelf at 0° longitude; given its location between a seasonal regime in the east and steady regime on the west, our framework (possibly due to the coarser 3° longitudinal bins) shows consistent seasonality in the heat (advective) transport but relatively steady melt rates. This could indicate a dominance of basal melt induced by currents rather than thermal driving (e.g., Gwyther et al. 2015)."*

**In this figure and the following figures, it would be helpful to have the shading of named regions as in Fig. 1. I recognize it's involved, but I think it would really assist interpretation.**

Thanks for your suggestion. We have included the grey shading referring to specific regions in Figure 2 and A1 (in the main manuscript).

**For the caption, please write out the TW definition as in Fig. 1.**

Thanks for your suggestion. We add the definition of TW in all figure captions where heat convergence in TeraWatts is shown (Fig. 2 and 3).

**Line 206-208: Does the model actually capture/represent coastal waves? If not, this shouldn't be listed as a driver of behaviour in the model. The other suggested processes are also lacking evidence to support their dominance, other than the high temporal variability, so it would be strengthened with more support.**

Thanks for your comment. We did mean coastal Rossby waves, which are resolved by our model. We modified this in the text. We acknowledge that we did not investigate in depth the processes playing a role in the high temporal variability of the heat convergences, as we think this is beyond the scope of the current manuscript and we plan to follow up on this in a future study. So, while we prefer to maintain this passage as it is, we appreciate the reviewer's comment.

**Line 214: The phrasing of advective heat convergence here is a bit misleading because the heat convergence can be both positive and negative (negative being essentially heat divergence). I agree that the advection term dominates the heat flux, but this sentence can be misinterpreted to mean that you see consistent heat convergence with a positive sign throughout the year, which isn't supported by Fig. 3, so I'd suggest rephrasing.**

Thanks for your comment. We agree with the reviewer and have modified the description of the advective heat transport in this passage, which now reads:

*"Although ADV plays an important role in warming episodes, it is also responsible for cooling events that occur mostly in winter months, also with high temporal variability.".*

**Line 226-228: What is the evidence for the proposed mechanism of spreading of wintertime cooled water? Fig. 4 is a very helpful figure overall, but it does not provide a very convincing link between the surface fluxes and the bottom temperature conditions and this is a key point in the results/discussion.**

Thanks for your question. We have included further evidence on these mechanisms, as answered above in the main points. In particular, we have now included a figure with cross-shelf transects in the front of the Totten and Getz ice shelves (Figure 6 in the main manuscript/Figure R4 on this document), where we show near-freezing temperatures during winter for the entire column in front of the Totten (and close to the Dalton Polynya, Fig. R4a). In contrast, the Getz ice front region shows no signal of wintertime convection.

**Fig. 4: It is hard to see differences in spatial distribution and magnitude of melt rates in this figure. To more clearly demonstrate this point, I'd suggest visualising seasonal melt rate anomaly in panels b/c, e/f and the mean melt rate in panel a.**

Thanks for your suggestion. We have added the melt anomalies in the panels of c and f of Figs. 4, 5, A7-15.

**Line 237-238: Is the dominance of mCDW below 300 m inconsistent with the overall argument of the difference between East and West Antarctic melt regimes? Please comment.**

Thanks for your question. This is a good point. The classification between seasonal and steady regimes used in Figure 1b accounts for the total melting, implying that both shallow (<300m) and deep (>300m) melting are accounted for when determining the melt regime classification. A seasonal ice shelf regime implies that the seasonal signal from shallow melting dominates over the steadier melting associated with deep melting. Among the analysed ice shelves, the ice shelves classified as steady regime (Sulzberger, Getz, PIG) show distinct contributions from both shallow and deep melting (Fig. 3f, A4g,h). Nevertheless, the steady regime ice shelves show no consistent

seasonality in the shallow melting. Similarly, the ice shelves classified as seasonal regime (Amery-West, Shackleton, Totten, Mertz, Ross) also present distinct ratios between shallow and deep melting (Fig. 3e, A5i, A6g,h,i), often with substantial deep melting. However, seasonal ice shelves have a clear impact from the seasonal warming, controlling mode 3 (shallow) melt variability. To make this point clearer, we amend the following passage in the discussion (lines 313-314):

*"At steady melting regimes, such as in the Getz region, the variability of the CDW-driven melt dominates, and the shallow melting does not present a consistent seasonality."*

**Line 250-255: The flow in this region is also strongly influenced by the cross-shelf density gradient, which sea ice and seasonal surface conditions do play a role in. So, the general idea of correlating polynya location with bottom temperatures captures one aspect of changes, but is certainly not comprehensive. I'd suggest adding a comment on other factors that may contribute.**

Thanks for the suggestion. We agree that cross-shelf density gradients could play a role in the continental shelf circulation in the Amundsen Sea and the seasonal anomalies seen in Fig. 5. We add some words on this at lines 283-284:

*"The seasonal variability of the surface heat flux and sea-ice conditions could play a role in the seasonal variations of the shelf circulation via cross-shelf density gradients (Li et al. 2025)."*

**Discussion and conclusions**

**Line 287-289: These sentences have some typos and reference formatting issues.**

Thanks for your comment; we have reworded the sentences in lines (now at lines XXX).

**Line 290-291: "giving confidence that the model captures relevant processes, such as the landfast ice effect on coastal polynya locations." The model specifically can't capture the actual processes, just the effect, because it does not have a sea ice model. Suggest removing or clarifying.**

Thanks for your suggestion. We reword this sentence to point out that the model represents some of the effects from these processes, which are to some extent included in the observational estimates used to force WAOM:

*"Coastal polynyas in this study are represented from observational estimates, giving confidence that the model captures, at least partially, the effects from relevant processes, such as the landfast ice effect on coastal polynya locations (Achter et al., 2022)."*

**Line 295: Does the seasonality of warm mCDW intrusions in Prydz Bay conflict with the argument of mode 3 melting suggested in this region? If so, I think it's important to more clearly identify this difference and comment on the reasons.**

Thanks for your comment. We have amended these sentences to reflect our results for Prydz Bay better. Although the deep melt induced by mCDW has a minor but consistent seasonality (Fig. A5i), the seasonality in shallow melting dominates the total melting:

*"In Prydz Bay, seasonality of warm intrusions has been identified from both observations and modelling (Galton-Fenzi et al., 2008; Gao et al., 2024). In our model, although deep melting driven by mCDW increases during winter, the overall basal melt seasonality is dominated by enhanced shallow melting in summer. This aligns with observational findings of summer-intensified melting (Gao et al., 2024)."*

**Line 300: "Most of this observed seasonality points to similar mechanisms as described". Please specify the similar mechanisms you are referring to.**

Thanks for the suggestion. We reformulated this sentence, which now reads:

*"Overall, most of this observed seasonality across these regions points to increased basal melting during the summer months, consistent with the shallow melting mechanism represented in our model."*

**Line 312-318: You might also want to consider the findings in Haigh & Holland (2024) for the role of sea ice on decadal variability in the Amundsen Sea in this portion of the discussion.**

Thanks for your suggestion. We have now cited this publication when referring to the mechanisms of decadal variability in the Amundsen Sea.

**Line 324: "The Bellingshausen Sea is the only region which holds substantial differences" Sudden transition from the previous paragraphs, consider linking to the previous paragraphs for continuity.**

Thanks for your suggestion. We have improved this paragraph in the discussion, which now reads:

*"Although the ice shelves in the Bellingshausen Sea were classified under the seasonal regime in the WAOM 4 km simulation, it is important to note that this region exhibits the most substantial differences across WAOM configurations."*

**Line 330: Need a link as to why you're mentioning the differences in steady versus seasonal melting regimes between model resolution and what it means for the findings. Does it mean that it's uncertain what the regime is or that high resolution scale processes dominate the seasonal behaviour etc.**

Thanks for the suggestion. We included a few more words on the implications of these differences across model resolution (lines 374-375):

*"This discrepancy in melt regime classification across WAOM resolutions highlights an important source of uncertainty in our results, stemming from model resolution and associated biases."*

**Line 331: "These results show how shallow". Specify which results you are referring to.**

Thanks for the comment. We rephrase the first sentence of this paragraph, which now reads:

*"Overall, the results shown in this study suggest an important role of the shallow melting as a source of melt variability at sub-annual timescales."*

**Line 353-354: Given the role of wind in CDW transport on the shelf in West Antarctica, could that be why this is the dominant mode in your study?**

Thanks for your comment. This is a good point. We have added a sentence on this hypothesis in the discussion (lines 405-406):

*"This high sensitivity could be contributing to melt variability being less dependent on the surface heat flux, in contrast to the East Antarctica sector."*

**Line 374: SAM introduction is a bit sudden since it isn't mentioned in the discussion. I'd suggest either bringing it up earlier or removing the mention here.**

Thanks for your suggestion. We replaced the mention to SAM with a more generic wording on climate variability:

*"Climate change-induced impacts on seasonality (Timmermann and Goeller, 2017; Mosbeux et al., 2023) and sea-ice regime shifts (Purich and Doddridge, 2023), in particular anomalous summertime conditions associated with climate and atmospheric circulation anomalies (Clem et al., 2024), can have potentially large implications for shallow melting and thus to Antarctic Ice Sheet mass loss and global sea level rise."*

**References**

**Dundas, V., Darelius, E., Daae, K., Steiger, N., Nakayama, Y., & Kim, T. W. (2022). Hydrography, circulation, and response to atmospheric forcing in the vicinity of the central Getz Ice Shelf, Amundsen Sea, Antarctica. Ocean Science, 18(5), 1339-1359.**

**Haigh, M., & Holland, P. R. (2024). Decadal variability of ice-shelf melting in the Amundsen Sea driven by sea-ice freshwater fluxes. Geophysical Research Letters, 51(9), e2024GL108406.**

**Khazendar, A., Schodlok, M. P., Fenty, I., Ligtenberg, S. R. M., Rignot, E., & Van Den Broeke, M. R. (2013). Observed thinning of Totten Glacier is linked to coastal polynya variability. Nature Communications, 4(1), 2857.**

Kind regards,
Fabio Boeira Dias and co-authors

References:

Li, Z., Wang, C., and Zhou, M. (2025): A Model Analysis of Circumpolar Deep Water Intrusions on the Continental Shelf Break in Amundsen Sea, Antarctica, Journal of Geophysical Research: Oceans, 130, https://doi.org/10.1029/2024JC022210.

Purich, A., & England, M. H. (2021). Historical and future projected warming of Antarctic Shelf Bottom Water in CMIP6 models. *Geophysical Research Letters*, 48, e2021GL092752. https://doi.org/10.1029/2021GL092752